# CTCF knockout in zebrafish induces alterations in regulatory landscapes and developmental gene expression

Martin Franke [1,3], Elisa De la Calle-Mustienes[1,3], Ana Neto[1], María Almuedo-Castillo [1], Ibai Irastorza-Azcarate[2], Rafael D. Acemel[1], Juan J. Tena [1], José M. Santos-Pereira [1✉] & José L. Gómez-Skarmeta[1,4]

Coordinated chromatin interactions between enhancers and promoters are critical for gene regulation. The architectural protein CTCF mediates chromatin looping and is enriched at the boundaries of topologically associating domains (TADs), which are sub-megabase chromatin structures. In vitro CTCF depletion leads to a loss of TADs but has only limited effects over gene expression, challenging the concept that CTCF-mediated chromatin structures are a fundamental requirement for gene regulation. However, how CTCF and a perturbed chromatin structure impacts gene expression during development remains poorly understood. Here we link the loss of CTCF and gene regulation during patterning and organogenesis in a *ctcf* knockout zebrafish model. CTCF absence leads to loss of chromatin structure and affects the expression of thousands of genes, including many developmental regulators. Our results demonstrate the essential role of CTCF in providing the structural context for enhancer-promoter interactions, thus regulating developmental genes.

[1] Centro Andaluz de Biología del Desarrollo (CABD), Consejo Superior de Investigaciones Científicas/Universidad Pablo de Olavide, Seville, Spain. [2] Berlin Institute for Medical Systems Biology, Max-Delbrück Center for Molecular Medicine, Berlin, Germany. [3] These authors contributed equally: Martin Franke, Elisa De la Calle-Mustienes. [4] Deceased: José L. Gómez-Skarmeta. ✉email: jmsanper1@upo.es

Vertebrate genomes are folded within the nucleus in a hierarchical manner leading to distinct levels of chromatin structure that range from chromosome territories to nucleosomes[1–5]. At the kilo- to megabases scale, chromatin is organized in topologically associating domains (TADs), which comprise genomic segments with higher interaction frequencies than with adjacent regions[6–12]. According to the current theory, TADs emerge when the cohesin complex, while extruding chromatin, is halted by the 11-zinc-finger DNA-binding protein CTCF[13,14]. CTCF is a transcriptional repressor and insulator that acts also as an architectural protein, mediating long-range chromatin looping and contributing to TAD insulation by demarcating its boundaries[15–20]. TADs facilitate the contact of *cis*-regulatory elements (CREs) with promoters located within them while preventing interactions with promoters located in neighboring TADs. Therefore, TADs have been suggested to be a fundamental requirement for gene regulation, but to what extent is still under debate[21–23]. Depletion of CTCF in mammalian in vitro systems causes only moderate transcriptional alterations[19,24,25], in agreement with some in vivo studies[26,27]. However, targeted deletions of several CTCF sites and subsequent quantitative gene expression analyses reveal significant loss of gene expression[28–30]. Moreover, genomic structural variations that rearrange TAD boundaries lead to enhancer-promoter rewiring, alterations in gene expression, and congenital malformations[31–35]. In fact, due to the essential function of CTCF during the cell cycle and the early embryonic lethality in mice[36–39], our understanding of its function in vivo during organogenesis is limited to a few physiological contexts.

Here, we analyze the genome-wide effect of CTCF in a zygotic *ctcf* knockout mutant in zebrafish that allows us to monitor the effect of CTCF loss of function during embryonic development. CTCF absence leads to loss of chromatin structure in zebrafish embryos and affects the expression of thousands of genes, including many developmental genes. The loss of insulation in the *ctcf* mutants decreases intra-TAD and increases inter-TAD chromatin interactions of active promoters, resulting in gene miss-expression. In addition, chromatin accessibility, both at CTCF binding sites and CREs is severely compromised in *ctcf* mutants, likely due to miss-expression of transcription factor (TF) genes. Probing chromatin interactions of developmental genes at high resolution, we further demonstrate that promoters fail to fully establish long-range contacts with their associated regulatory landscapes (RLs), while 3D modeling of these loci reveals increased enhancer–promoter distances in the absence of CTCF. As a consequence, we observe altered gene expression patterns and disruption of developmental programs. Our results demonstrate that CTCF is essential to regulate gene expression during embryonic development at multiple levels, including the constraining of enhancer–promoter interactions within developmental RLs.

## Results

**Generation of a zebrafish *ctcf*$^{-/-}$ zygotic mutant**. In order to study the requirement of CTCF in an in vivo vertebrate model, we have generated a *ctcf* zygotic knockout mutant in zebrafish. Using CRISPR/Cas9 with two single guide RNAs (sgRNAs), we obtained heterozygous *ctcf*$^{+/-}$ adult individuals carrying a 260-bp deletion, encompassing exons 3 and 4 of the *ctcf* gene that leads to a premature stop codon within exon 4. While *Ctcf*$^{-/-}$ zygotic knockout mice die at peri-implantation stages[36], zebrafish *ctcf*$^{-/-}$ mutants undergo gastrulation and organogenesis and develop normally until pharyngula stages, around 24 h-post-fertilization (hpf). At 24 hpf, *ctcf*$^{-/-}$ mutants are phenotypically indistinguishable from their heterozygous and wild-type siblings.

However, at 48 hpf, *ctcf*$^{-/-}$ embryos showed a clear phenotype that included pigmentation defects, heart edema and reduced size of head and eyes, dying shortly after this stage. In contrast, heterozygous *ctcf*$^{+/-}$ embryos show a wild-type phenotype (Fig. 1a). The expected N-terminal truncated CTCF protein would be depleted of all zinc finger domains but one, preventing CTCF binding to chromatin. Western blot analysis at 48 hpf with an antibody recognizing the N-terminal end of CTCF showed similar levels of wild-type protein in *ctcf*$^{+/-}$ and complete loss in *ctcf*$^{-/-}$ embryos (Supplementary Fig. 1a). The absence of a truncated protein from the mutant allele in *ctcf*$^{+/-}$ and *ctcf*$^{-/-}$ suggested degradation of the mutant CTCF protein. We then determined CTCF chromatin binding events in wild type at 24 and 48 hpf by ChIPmentation, using a zebrafish specific antibody[40] (Supplementary Fig. 1b, c). We identified around 30,000 sites at both stages, with most of them (17,160) being stably bound by CTCF. Motif analysis of stable- and stage-specific binding sites revealed CTCF as the most enriched motif. In 48 hpf *ctcf*$^{-/-}$ mutant embryos, except for a small subset (848) of strong binding sites that persisted in agreement with previous observations[41], the majority of sites were depleted of CTCF, further supporting the absence of chromatin binding events.

Then, we hypothesized that the delayed lethality in zebrafish *ctcf*$^{-/-}$ mutants could be due to a prolonged maternal contribution of CTCF protein during early embryonic development. To quantify the maternally provided CTCF in our zebrafish model, we performed CTCF immunofluorescence analyses during early development in wild-type, *ctcf*$^{+/-}$ and *ctcf*$^{-/-}$ siblings (Fig. 1b). Maternal *ctcf* mRNA is detected at least until 75% of epiboly (8 hpf, gastrulation)[42] and we observed significantly reduced CTCF protein levels at 80% of epiboly in *ctcf*$^{-/-}$ mutant embryos. CTCF protein was also detected at 18 somites stage (18 hpf), suggesting prolonged CTCF stability. However, nuclear staining of maternal CTCF at 24 hpf was no longer detected in *ctcf*$^{-/-}$ mutant embryos. Anterior and posterior embryonic domains showed a similar dynamic of CTCF loss at these late stages with only unspecific background levels remaining at 24 hpf (Supplementary Fig. 1d). In contrast, *ctcf*$^{+/-}$ embryos maintain CTCF protein levels that are only slightly reduced when compared to their wild-type siblings. This indicates that the delayed lethality of zebrafish *ctcf*$^{-/-}$ mutants compared to mice is due to the presence of maternal CTCF protein for a longer time, at least until the 18 somites stage in zebrafish embryonic development. Therefore, our *ctcf*$^{-/-}$ zebrafish mutant provides a unique tool to examine the contribution of this protein in genome architecture, gene expression, and body plan formation in a vertebrate model system using a combination of chromosome conformation capture, transcriptomic, and epigenomic techniques.

**CTCF is required for chromatin organization in zebrafish**. We first analyzed whether the absence of CTCF in zebrafish embryos caused loss of chromatin structure, as previously reported in in vitro models[19,24,25]. For this, we performed HiC experiments in wild-type and *ctcf*$^{-/-}$ whole embryos at 24 and 48 hpf and visualized the data at 10-kb resolution. Similar to previous reports[43,44], we found that chromatin domain were established at 24 hpf in wild-type embryos detecting 2,599 and 2,438 TADs at 24 and 48 hpf, respectively, based on insulation scores[45]. We observed only minor differences between the wild-type stages with slightly stronger TAD insulation at 48 hpf (Fig. 1c; Supplementary Fig. 2a–d). Other 3D chromatin features commonly detected at this scale, such as loops and stripes, were also observed. In contrast, we found a general loss of chromatin structure in *ctcf*$^{-/-}$ embryos at both stages, characterized by

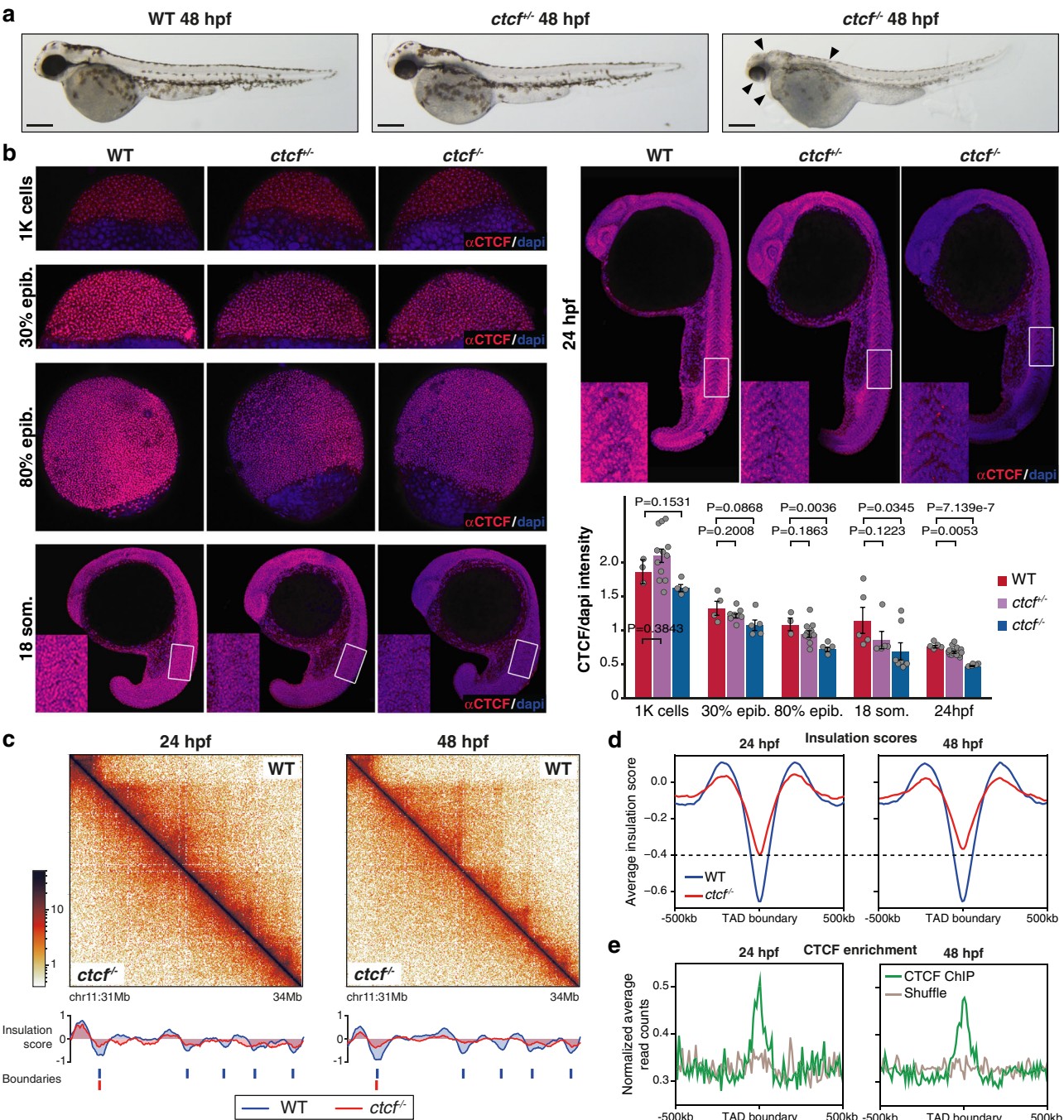

**Fig. 1 Knockout of *ctcf* in zebrafish embryos disrupts chromatin structure. a** Pictures of wild-type (WT), *ctcf*[+/−] and *ctcf*[−/−] zebrafish embryos at 48 h post fertilization (hpf) showing homozygous mutant phenotypes, including the reduced size of head and eyes, heart edema, and defective pigmentation (arrow heads). Scale bars represent 250 μm. **b** Whole-mount embryo immunofluorescence of CTCF (red) and dapi (blue) in WT, *ctcf*[+/−] and *ctcf*[−/−] zebrafish embryos at the stages of 1000 cells (1 K cells), 30% of epiboly (30% epib.), 80% of epiboly (80% epib.), 18 somites (18 som.) and 24 hpf showing the maternal contribution of CTCF protein. Relative quantification of CTCF/dapi signal with average values ± standard error is shown. Statistical significance was measured using a two-sided Student's *t* test. The number of embryos for WT, *ctcf*[+/−] and *ctcf*[−/−] used for quantification are as follows: 1 K cells (*n* = 3, *n* = 12, *n* = 3); 30% epib. (*n* = 4, *n* = 8, *n* = 5); 80% epib. (*n* = 3, *n* = 10, *n* = 5); 18 som. (*n* = 5, *n* = 5, *n* = 7) and 24 hpf (*n* = 5, *n* = 17, *n* = 5). Source data are provided as a Source Data file. **c** HiC normalized contact maps at 10-kb resolution from WT and *ctcf*[−/−] zebrafish embryos at 24 and 48 hpf. A 3-Mb genomic region in chr11 is plotted, aligned with the insulation scores and the called topologically associating domain (TAD) boundaries. **d** Average insulation score profiles of WT and *ctcf*[−/−] zebrafish embryos at 24 and 48 hpf around the TAD boundaries called in the WT. **e** Average CTCF ChIP-seq signal around TAD boundaries (green) and a shuffle control (brown) in WT embryos at 24 and 48 hpf.

reduction of intra-TAD contacts and TAD insulation (Fig. 1c, d; Supplementary Fig. 2a–e). Interestingly, the loss of TAD insulation was slightly stronger at 48 than at 24 hpf (Fig. 1d). This is in agreement with the observed gradual decay of maternally

provided CTCF and possibly indicates that some residual CTCF protein at 24 hpf still maintained some functions. TAD boundary calling detected 2,582 and 2,416 boundaries in wild-type embryos at 24 and 48 hpf, respectively, while only 1666 and 1157

boundaries were called in $ctcf^{-/-}$ embryos at the corresponding stages (Supplementary Fig. 2f). Mutant-specific boundaries were less likely to contain CTCF binding sites and showed a weaker insulation score. In addition, we detected higher RNA levels at mutant-specific and common boundaries (Supplementary Fig. 2g, h), suggesting that transcription might sustain boundary formation in the absence of CTCF. These data confirmed that CTCF is essential for 3D chromosome organization in zebrafish embryos, as described for other vertebrates including mammals and frogs[19,24,46].

Next, we analyzed A and B compartments in wild-type and $ctcf^{-/-}$ embryos and, although we found a similar distribution of compartments, we detected a less intense plaid pattern of the Pearson's correlation matrices in $ctcf^{-/-}$ embryos both at 24 and 48 hpf (Supplementary Fig. 3a). Saddle plots of the interaction enrichment of genomic bins sorted by their eigenvalues detected a slight but consistent decrease in compartmentalization in mutant embryos that affected specifically interactions between active regions (Supplementary Fig. 3b–d). This is in agreement with recent CTCF depletion experiments in mammalian cells[47] and suggests that CTCF may be required for higher-order chromatin structure.

We then analyzed CTCF binding at TAD boundaries and found that CTCF was indeed enriched at these regions (Fig. 1e), with 46–48% of wild-type boundaries containing high-confidence CTCF binding sites (Supplementary Fig. 4a, b). In addition, the consensus motifs of CTCF at these binding sites around TAD boundaries were preferentially located in a convergent orientation (Supplementary Fig. 4c), consistent with the previous observations[6,10,43,48,49]. This enrichment was also observed using previously published HiC data of 24-hpf zebrafish embryos[43], but not with recently reported ChIP-seq data using an HA-tagged CTCF protein in zebrafish[50] (Supplementary Fig. 4d). In contrast to the latter study, we detected a clear CTCF motif prevalence and a high enrichment of CTCF at TAD boundaries in zebrafish, using an antibody against endogenous CTCF[51]. Next, we called chromatin loops in wild-type embryos and detected 1,436 and 1,297 loops at 24 and 48 hpf, respectively, 40% of which contained CTCF binding sites at both loop anchors (Supplementary Fig. 4e). Interestingly, aggregate peak analysis of CTCF-containing chromatin loops showed a marked decrease in intensity in $ctcf^{-/-}$ mutants, while loops without CTCF were less affected by CTCF loss (Supplementary Fig. 4e), suggesting that they may be sustained by CTCF-independent mechanisms. Therefore, we conclude that CTCF is essential for the establishment of most chromatin loops in zebrafish embryos, similarly to other vertebrates[19,46].

**Developmental gene expression requires CTCF.** To analyze the effects of CTCF absence over gene expression in vivo, we performed RNA-seq on whole embryos at 24 and 48 hpf. At 24 hpf, we detected 260 upregulated and 458 downregulated genes (Fig. 2a). However, at 48 hpf, we detected as many as 2,730 upregulated and 3,324 downregulated genes (Fig. 2b). Strikingly, while differentially expressed genes (DEGs) at 24 hpf were enriched only in biological functions related to immune and DNA damage responses, DEGs at 48 hpf were enriched, among other general functions, in transcription regulation and developmental processes including skeletal muscle development or nervous system development (Fig. 2c, d). Compared to in vitro CTCF depletion approaches[19,24,25], our results reveal a considerable impact on developmental gene expression and indicate that CTCF is required for the expression of developmental genes during zebrafish embryogenesis. Next, we analyzed gene expression changes in the transition from 24 to 48 hpf in wild-type embryos and found that genes that get activated in this period tend to be downregulated in $ctcf^{-/-}$ embryos, and vice versa

(Supplementary Fig. 5a–d), indicating that many developmental genes fail to acquire their normal expression levels during this developmental period.

We then explored the possible function of CTCF to directly regulate DEGs by analyzing its binding to their transcription start sites (TSSs). At 24 hpf, we found a clear bias of CTCF binding toward the TSS of downregulated genes (68.8%) as compared to upregulated genes (12.3%) (Fig. 2e, f). This confirms previous observations[19,24] and suggests distinct mechanisms of CTCF function at activated and repressed genes. At 48 hpf, only 36% of downregulated and 17.3% of upregulated genes showed CTCF binding at their TSSs (Fig. 2e, f). In contrast to previous data[19], we did not detect any clear bias in the orientation of the CTCF motif at TSS relative to transcription (Fig. 2g). Finally, we observed that downregulated genes that are enriched in developmental functions were mainly those without CTCF bound at their TSSs (Supplementary Fig. 5e), raising the possibility that developmental genes could be miss-regulated either indirectly by miss-regulation of downstream CTCF targets or due to defects in chromatin folding. Altogether, these data show that CTCF absence leads to altered developmental gene expression that may account for the observed developmental abnormalities.

**CTCF-mediated insulation restricts promoter contacts within TADs.** To address changes in long-range gene regulation that could be associated with loss of CTCF-mediated chromatin structure, we profiled genome-wide interactions of active promoters using HiChIP[52] in wild-type and $ctcf^{-/-}$ embryos at 48 hpf, pulling down chromatin with an anti-H3K4me3 antibody. We used FitHiChIP to calculate differential loops between both conditions[53]. To avoid confusion with loops called in HiC data, we will refer to loops called in HiChIP data as 'HiChIP loops' hereafter. Unambiguous assignment of differential chromatin contacts using this technique requires the analysis of promoter-centered interactions showing similar coverage levels of H3K4me3 in wild-type and $ctcf^{-/-}$ mutants. We detected 64,145 HiChIP loops from 13,083 gene promoters that fulfilled this requirement, from which 417 loops (from 591 genes) showed increased, and 261 loops (from 262 genes) decreased chromatin contacts in mutant embryos with high confidence levels (Fig. 3a). The obtained number of differential HiChIP loops is in accordance with those reported in a recent study in mouse ESCs and neural progenitor cells[25]. Aggregate peak analysis confirmed differential HiChIP loop intensity, and we also observed an average decrease for the stable loops (Fig. 3b), suggesting a general negative impact of CTCF loss on promoter contacts.

Next, we asked whether differential HiChIP loops could be directly related to the loss of TAD insulation and boundaries. For this, we first analyzed the genomic distances established by the differential HiChIP loop categories and found that increased loops spanned significantly longer distances than stable loops, while decreased loops spanned shorter distances (Fig. 3c). We further observed that 57.8% of increased loops crossed wild-type TAD boundaries, which is a significantly higher proportion than for stable loops (35.8%; Fig. 3d) and suggestive of an increase in inter-TAD interactions upon CTCF loss. The opposite trend was observed for decreased loops where only 4.2% of loops crossed TAD boundaries, indicating a reduction of intra-TAD interactions (Fig. 3d). Instead, the decreased HiChIP loop category showed a higher enrichment for CTCF binding at both loop anchors (32.2% of loops; Fig. 3e) and a higher overlap with HiC loops (29.9% of loops; Supplementary Fig. 6a), confirming the reduction of CTCF-mediated chromatin contacts within TADs. Interestingly, 67.8% of decreased HiChIP loops did not show CTCF binding at both anchors, suggesting that the absence of

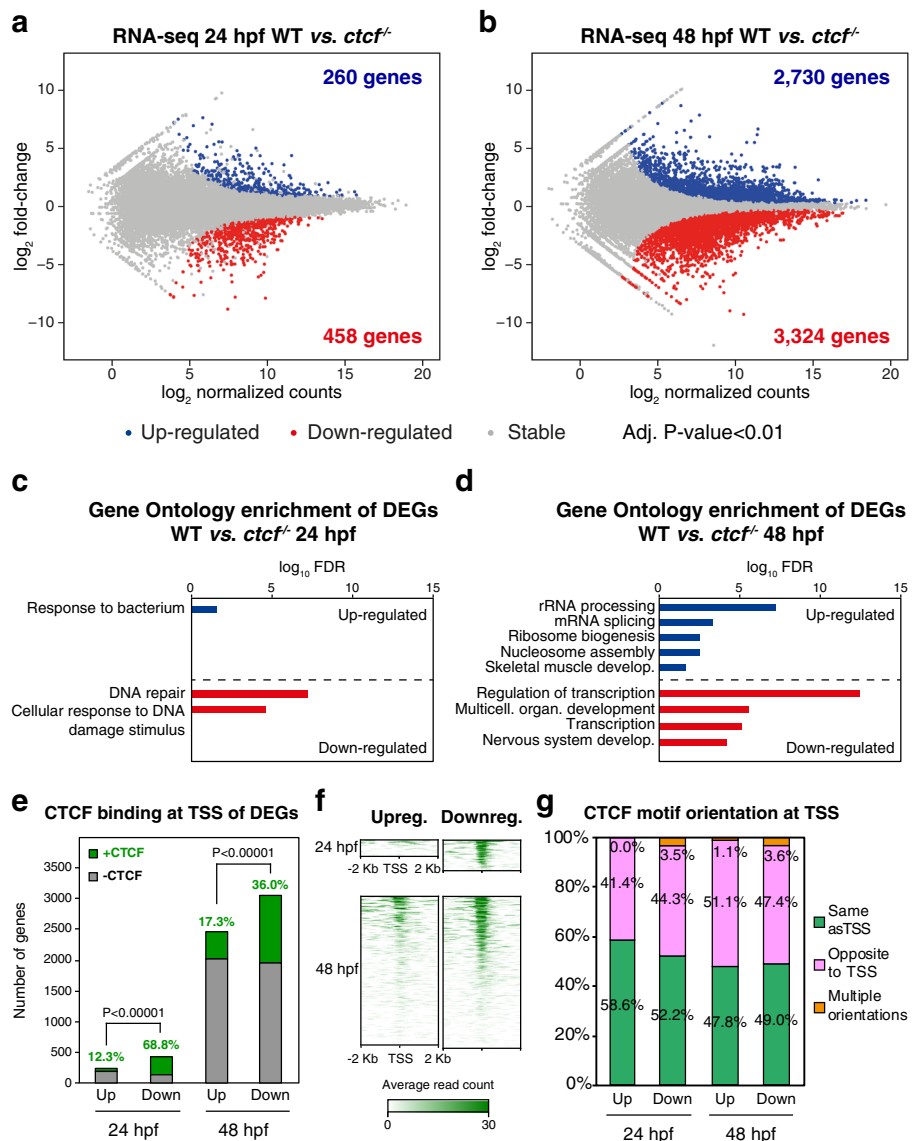

**Fig. 2 CTCF absence in zebrafish embryos leads to altered developmental gene expression. a, b** Differential analyses of gene expression between WT and *ctcf*[−/−] embryos at 24 (**a**) and 48 hpf (**b**) from RNA-seq data (*n* = 2 biological replicates per condition). The log$_2$ normalized read counts of WT transcripts versus the log$_2$ fold-change of expression are plotted. Transcripts showing a statistically significant differential expression (adjusted *P*-value < 0.01) are highlighted in blue (upregulated) or red (downregulated). The number of genes that correspond to the upregulated and downregulated transcripts are shown inside the boxes. **c, d** Gene Ontology (GO) enrichment analyses of biological processes for differentially expressed genes (DEGs) in *ctcf*[−/−] embryos at 24 (**c**) and 48 hpf (**d**). Terms with a false discovery rate (FDR) < 0.05 are shown and considered as enriched. **e** Number of differentially expressed genes (DEGs) at 24 and 48 hpf showing (green) CTCF binding at their transcription start sites (TSS) or not (gray). **f** Heatmaps showing CTCF ChIP-seq signal around the TSS of DEGs at 24 and 48 hpf. **g** Percentage of CTCF motif orientation at ChIP-seq peaks overlapping TSS and relative to transcriptional orientation.

CTCF-mediated insulation decreases the contact probability within TADs independently of CTCF binding. Altogether, these data suggest that CTCF function in TAD insulation contributes to intra-TAD and prevents inter-TAD promoter contacts.

Finally, we assessed whether differential HiChIP loops were associated with changes in gene expression. We found that increased loops were indeed significantly more associated with the TSS of upregulated genes than stable loops, while decreased loops showed a higher overlap with the TSS of downregulated genes (Fig. 3f), and in particular with TSSs bound by CTCF (Supplementary Fig. 6b). This association was further supported by a stronger average fold-change for DEGs overlapping the respective differential loop categories (Fig. 3g), suggesting that the altered promoter interactions contribute to transcriptional regulation.

Among the DEGs with altered promoter interactions, we found a further enrichment of transcriptional regulation and developmental functions when compared to all DEGs (Supplementary Fig. 6c), indicating that developmental genes are more likely to be affected by changes in chromatin topology induced by CTCF loss. For example, the extended locus of the *smad7* gene, which is downregulated in *ctcf*[−/−] embryos, contains two decreased HiChIP loops involving the *smad7* promoter and within their respective TAD. At this locus, we also detected a long-range HiChIP loop with increased contact frequency connecting the promoter of the up-regulated gene *tor1* with another TAD several hundred kb upstream (Fig. 3h). Altogether, these data indicate that CTCF contributes to reinforce and restrict promoter interactions within TADs and, thus, ensuring correct developmental gene expression.

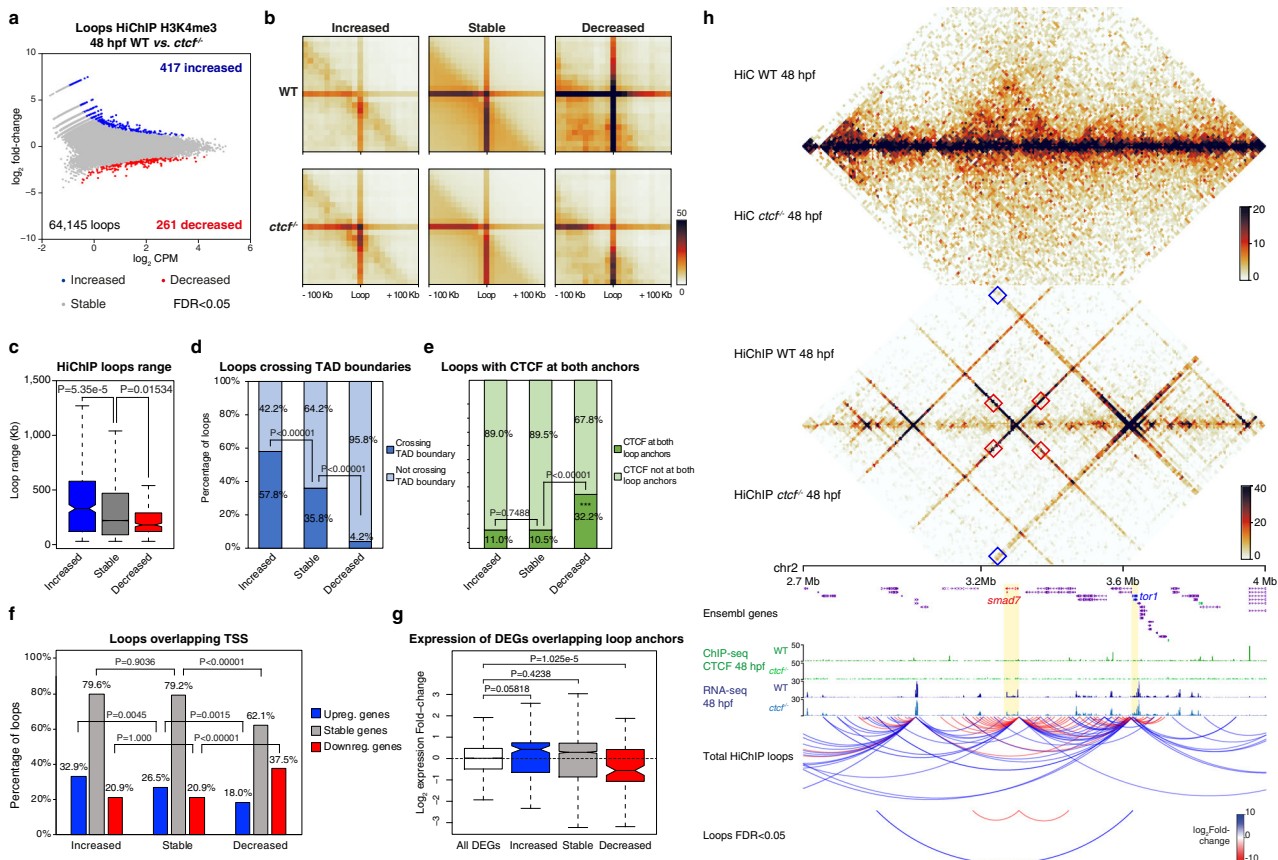

**Fig. 3 Differential promoter looping is associated with gene miss-expression. a** Differential analysis of H3K4me3 HiChIP loops between WT and *ctcf*$^{-/-}$ embryos at 48 hpf (*n* = 2 biological replicates per genotype) at 10-kb resolution. The log$_2$ normalized counts per million (CPM) of WT reads versus the log$_2$ fold-change of expression are plotted. Loops showing a statistically significant differential intensity (FDR < 0.05) are highlighted in blue (increased) or red (decreased). **b** Aggregate peak analysis centered at HiChIP loops for increased, stable and decreased loops in WT and *ctcf*$^{-/-}$ embryos. **c** Boxplots showing the distance between loop anchors (loop range) for increased, stable, and decreased loops. **d** Percentage of loops crossing WT topologically associating domain (TAD) boundaries for increased, stable, and decreased loop categories. **e** Percentage of loops showing CTCF binding at both anchors for increased, stable, and decreased loops. **f** Percentage of loops overlapping with the TSS of upregulated, stable, or downregulated genes for the respective loop categories. **g** Box plots showing the expression fold-change in *ctcf*$^{-/-}$ embryos at 48 hpf of all DEGs and those associated with increased, stable, and decreased loops. **h,** From top to bottom, heatmaps showing HiC and H3K4me3 HiChIP signal, tracks with CTCF ChIP-seq and RNA-seq, total analyzed HiChIP loops and differential loops (FDR < 0.05), for WT and *ctcf*$^{-/-}$ embryos at 48 hpf in a 1.3-Mb region of chromosome 2 containing the downregulated gene *smad7* and the upregulated gene *tor1*. Boxplots in **c** and **g** show center line, median; box limits, upper and lower quartiles; whiskers, 1.5× interquartile range; notches, 95% confidence interval of the median. Statistical significance was assessed using a two-sided Wilcoxon's rank-sum test in (**c**) and (**g**), and with a two-sided Fisher's exact test in (**d**–**f**).

**CTCF is required for chromatin accessibility at developmental CREs.** The expression of developmental genes is often regulated by multiple tissue-specific CREs, on which combinations of TFs are bound, giving rise to precise spatial and temporal expression patterns. Since CTCF absence affects the expression of developmental genes mostly without direct binding to their promoters, we reasoned that this could be due to miss-function of their associated CREs in addition to the observed alterations in enhancer–promoter interactions. To test this, we performed ATAC-seq in wild-type and *ctcf*$^{-/-}$ embryos at 24 and 48 hpf. At 24 hpf, we only found 56 differentially accessible regions (DARs), 21 with increased and 35 with decreased accessibility (Fig. 4a). However, at 48 hpf we found a total of 18,744 DARs, most of them decreased (18,138 sites vs. 606 increased) (Fig. 4b), temporally coinciding with the detected altered expression of developmental genes (Fig. 2b). Indeed, when we analyzed CREs gaining or losing accessibility in wild-type embryos from 24 to 48 hpf, we found that these sites failed to gain or lose accessibility in *ctcf*$^{-/-}$ embryos (Supplementary Fig. 7a–c), indicating that loss of CTCF impacts chromatin accessibility of thousands of CREs.

Motif enrichment analysis showed that the CTCF consensus binding sequence was specifically enriched in peaks with decreased accessibility, both at 24 and 48 hpf (Fig. 4c, d). We confirmed this by analyzing CTCF binding to DARs at 48 hpf and found that 7.6% of peaks with increased accessibility but 40.3% with decreased accessibility were bound by CTCF (Fig. 4e). Reduced DARs without CTCF binding did not show enrichment of the CTCF motif and tended to be more distally located to the nearest TSS (Supplementary Fig. 7d, e). Interestingly, we found that reduced DARs at 48 hpf, especially those bound by CTCF, showed slightly reduced accessibility already at 24 hpf (Fig. 4e). This suggests that the effect of CTCF loss on chromatin accessibility is progressive but less dynamic than the effect on chromatin structure at 24 hpf (Fig. 1c, d).

We also noticed that increased DARs, and in particular, those not bound by CTCF were enriched for the p53 family motif at 48 hpf. At this stage we also found increased expression of *tp53* and well-known p53 target genes (Fig. 4d; Supplementary Fig. 7d and Supplementary Fig. 8), pointing towards an increased apoptotic response in *ctcf*$^{-/-}$ mutants[36]. To test a possible contribution of

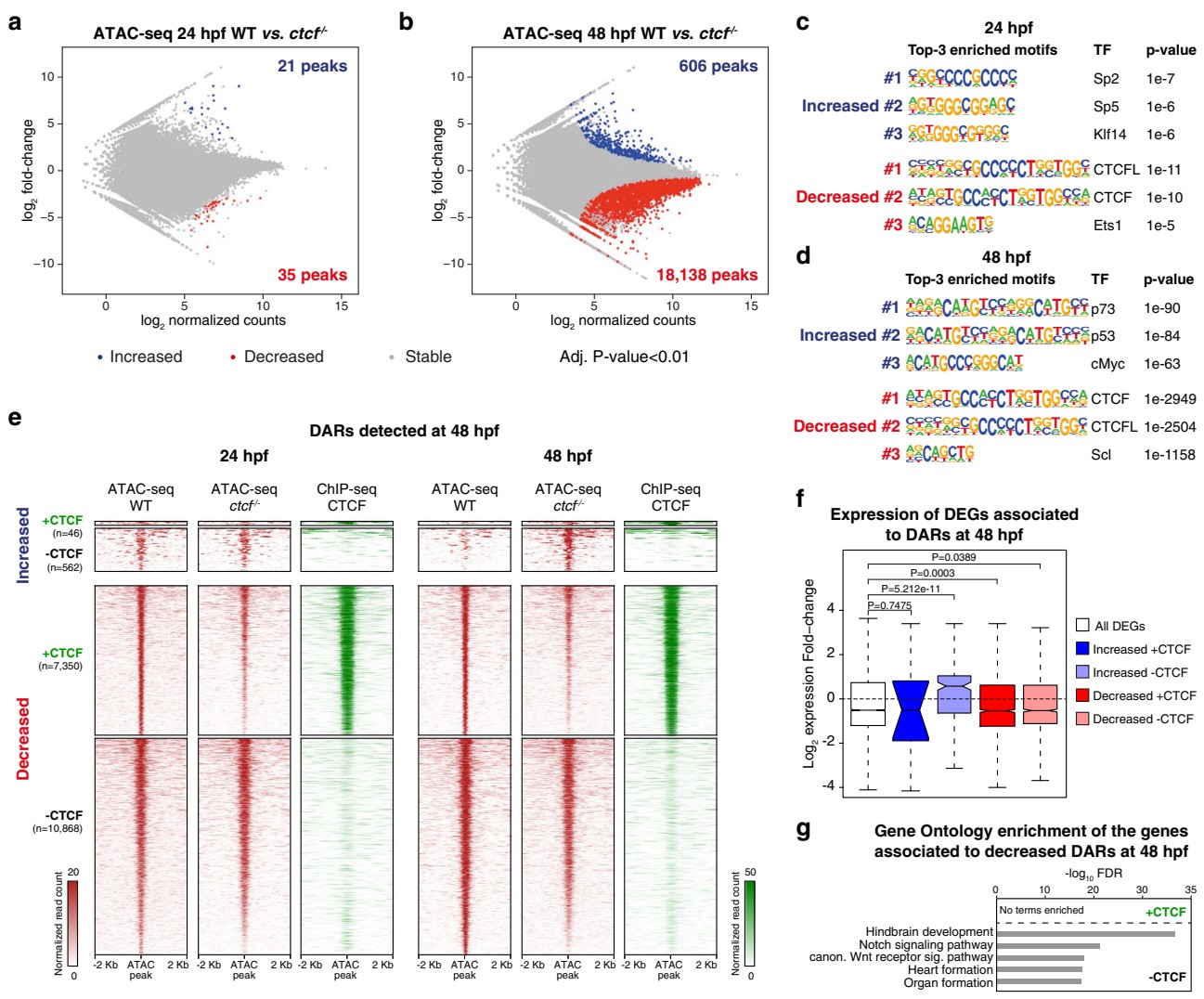

**Fig. 4 CTCF promotes chromatin accessibility at developmental *cis*-regulatory elements. a, b** Differential analyses of chromatin accessibility between WT and *ctcf*$^{-/-}$ embryos at 24 (**a**) and 48 hpf (**b**) from ATAC-seq data (*n* = 2 biological replicates per condition). The log$_2$ normalized read counts of WT ATAC peaks versus the log$_2$ fold-change of accessibility are plotted. Regions showing statistically significant differential accessibility (adjusted *P*-value < 0.01) are highlighted in blue (increased) or red (decreased). The total number of differential peaks is shown inside the boxes. **c, d** Motif enrichment analyses for the increased and decreased ATAC peaks in *ctcf*$^{-/-}$ embryos at 24 (**c**) and 48 hpf (**d**). The three motifs with the lowest p-values are shown for each case. **e** Heatmaps plotting normalized ATAC-seq signal in WT and *ctcf*$^{-/-}$ embryos (red), as well as CTCF ChIP-seq signal (green) at 48 hpf, for the differentially accessible regions (DARs) from (**b**) overlapping or not with CTCF sites. **f** Box plots showing the expression fold-change in *ctcf*$^{-/-}$ embryos at 48 hpf of all DEGs or only those associated with increased or decreased DARs, overlapping or not with CTCF sites. Boxplots represent the centerline, median; box limits, upper and lower quartiles; whiskers, 1.5× interquartile range; notches, 95% confidence interval of the median. Statistical significance was assessed using a two-sided Wilcoxon's rank-sum test. **g** GO enrichment analyses of biological processes for the genes associated with decreased DARs in *ctcf*$^{-/-}$ embryos at 48 hpf, overlapping or not with CTCF sites. GO terms showing an FDR < 0.05 are considered enriched.

p53 to the mutant phenotypes, we injected one-cell stage embryos with a morpholino to knock down *tp53* expression. We detected a reduction of p53-target gene expression and loss of p53-target motif in DARs with increased accessibility in morpholino-injected mutants, but global differential accessibility remained unaffected (Supplementary Fig. 8). Furthermore, the p53 knock-down did not change the mutant phenotype at 48 hpf, indicating that the phenotypic response may not be driven by pro-apoptotic processes.

Next, we associated DARs with nearby DEGs and found that the average change in gene expression was consistent with the tendency of changes in chromatin accessibility and independent of CTCF binding (Fig. 4f). Interestingly, only decreased DARs without CTCF binding were associated with genes enriched in developmental functions, such as hindbrain development or heart

formation (Fig. 4g). This indicates that loss of CTCF affects indirectly the accessibility of developmental CREs. We also noted that decreased DARs without CTCF binding sites were highly clustered within the RLs of developmental genes, many of them strongly reduced in the mutant (Supplementary Fig. 9a–c). This is consistent with the view that developmental genes frequently locate within large gene deserts containing many CREs. Indeed, we found that TADs containing miss-regulated developmental genes were larger and had more associated CREs than those containing non-developmental genes (Fig. 5a, b). Several examples illustrate this tendency. The *sall1a* gene, encoding a transcriptional repressor involved in organogenesis, is in a TAD whose structure was lost in *ctcf*$^{-/-}$ embryos (Fig. 5c). The expression of *sall1a* was reduced in the absence of CTCF and several CREs exhibited reduced accessibility with most of them,

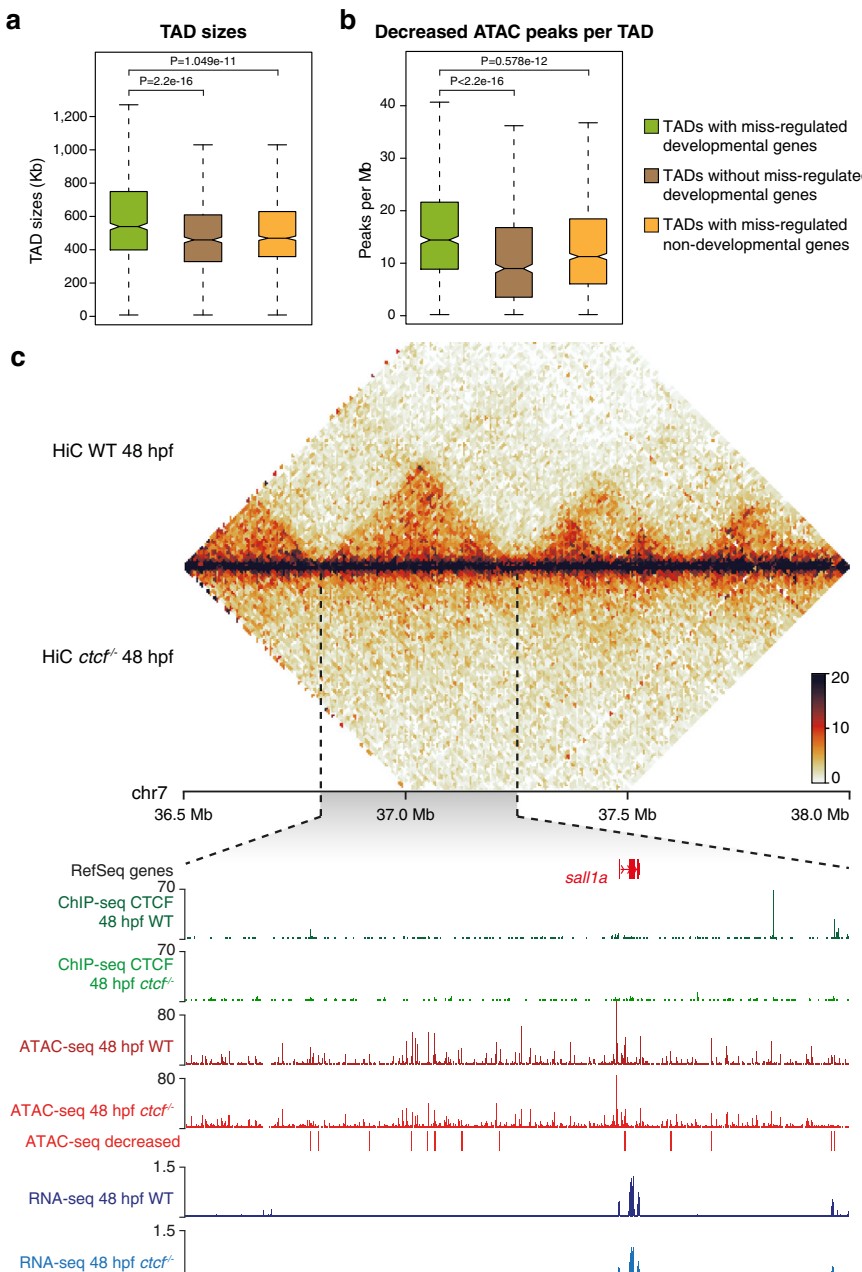

**Fig. 5 Clusters of *cis*-regulatory elements around developmental genes lose accessibility upon CTCF loss. a**, **b** Box plots showing the TAD sizes (h) and the number of decreased DARs per Mb (i) for TADs containing developmental miss-regulated genes, TADs not containing developmental miss-regulated genes, and TADs containing only non-developmental miss-regulated genes. Boxplots represent the centerline, median; box limits, upper and lower quartiles; whiskers, 1.5× interquartile range; notches, 95% confidence interval of the median. Statistical significance was assessed using a two-sided Wilcoxon's rank-sum test. **c** Top, heatmaps showing HiC signal in WT and *ctcf*$^{-/-}$ embryos at 48 hpf in a 1.5-Mb region of chromosome 7. Bottom, zoom within the *sall1a* TAD showing tracks with CTCF ChIP-seq, ATAC-seq, and RNA-seq at 48 hpf in WT and *ctcf*$^{-/-}$ embryos, as well as decreased ATAC-seq peaks. The downregulated *sall1a* gene is shown in red.

not binding CTCF. Other examples included the *lhx1a* and *sox11b* genes, both encoding developmental TFs (Supplementary Fig. 9d, e). Altogether, these data show that CTCF is required for the accessibility of thousands of CREs, many of which are associated with developmental genes.

Reduced accessibility at developmental CREs may be explained by reduced TF binding due to down-regulated TF gene expression. To test this possibility, we first selected TF families whose DNA binding motifs were enriched in DARs. We then plotted the expression fold-change of all transcripts associated with these families and with curated expression in zebrafish at 48 hpf. For TF

families associated with increased DARs, we found that only transcripts of the tp53 family were significantly upregulated in *ctcf*$^{-/-}$ embryos (Supplementary Fig. 10). In contrast, transcripts of all families associated with reduced DARs, including C2H2-type zinc fingers, bHLH, HMG, and homeobox, showed significantly reduced expression levels (Supplementary Fig. 10). Thus, the decreased accessibility at developmental CREs may be explained by a downregulation of TF genes caused by CTCF absence, which in turn would lead indirectly to miss-regulation of their downstream target genes. To quantify the extent of such indirect effects, we leveraged our ATAC-seq data to perform differential TF

binding analysis using a recently reported approach based on TF DNA footprints[54]. As expected, we detected the CTCF motif with the most reduced footprint signature, indicating reduced chromatin binding, but also 26 other motifs with altered footprints, 9 with increased and 15 with decreased footprints (Supplementary Fig. 11a–d). Next, we built a TF network based on the presence of TF footprints at TF gene promoters, using CTCF as a starting point and the motifs with differential footprints. This defined a CTCF TF network including 24 of the 26 motifs with differential footprints, which corresponded to 38 zebrafish orthologous genes (Supplementary Fig. 11e). Although 17 of them were miss-regulated in $ctcf^{-/-}$ embryos, our RNA-seq detected 452 further TF genes differentially expressed, indicating that only a small subset of miss-expressed TF genes could be explained by this CTCF network. Furthermore, the assignment of putative target genes to ATAC peaks containing these motifs identified only 47.9% of DEGs in our mutant (2895 out of 6049 genes) (Supplementary Fig. 11f). This analysis suggests that the role of CTCF as a transcriptional regulator and the potentially associated downstream TFs binding to DARs can explain a subset of the DEGs in $ctcf$ knockout mutants.

**CTCF is required for spatiotemporal expression patterns**. We have shown so far that CTCF is not only required for chromosome folding during zebrafish development, but also for the robust expression levels of many developmental genes and for their promoter interactions. To better assess gene miss-expression in relation to the loss of chromatin structure, we first investigated chromatin interactions at the enhancer-promoter level with high resolution by performing UMI-4C experiments[55]. We used developmental gene promoters as viewpoints to analyze their RLs in wild-type and $ctcf^{-/-}$ embryos at 48 hpf, such as the $ptch2$ promoter. $Ptch2$ is a patterning gene that encodes a cell receptor binding the Shh morphogen and whose expression was detected as upregulated by RNA-seq in $ctcf^{-/-}$ mutants (Fig. 6a). We found that contacts from the $ptch2$ promoter spanned a region of about 500 kb in wild-type embryos, establishing contacts with many genomic regions that included ATAC peaks (potential CREs) with and without CTCF binding (Fig. 6a). However, contacts within the $ptch2$ RL were drastically reduced in $ctcf^{-/-}$ embryos. The interaction profile was generally characterized by loss of long-range contacts but maintenance of some contacts at shorter ranges (Fig. 6a), consistent with observations in mammalian cells[25,56]. Genomic regions showing a reduced contact frequency with the $ptch2$ promoter included CTCF-binding sites as well as ATAC peaks not bound by CTCF. While some of those peaks losing contacts showed reduced accessibility in the mutant, others were not affected by CTCF absence, suggesting that the decreased enhancer-promoter interactions do not occur as a consequence of decreased TF binding. These results indicate that enhancer-promoter contacts were severely affected by the absence of CTCF, and in particular, long-range interactions.

Next, we investigated whether this loss of contacts altered the expression pattern of $ptch2$ by performing the whole embryo in situ hybridization. $Ptch2$ mRNA was detected in the brain, pharyngeal arches, and pectoral fin buds of wild-type embryos, but we found that this pattern was severely altered in $ctcf^{-/-}$ embryos (Fig. 6b). Consistent with the upregulation in our bulk RNA-seq data, $ptch2$ expression in mutant embryos was extended to broader regions of the brain, pharyngeal arches, neural tube, and a prominent expansion of expression was observed in the somites. The expression of $ptch2$ in the pectoral fin buds was not detected likely due to their severely impaired development at this developmental stage (Fig. 6b), corroborating previous results in $ctcf$-deficient mouse limb buds[39]. $Ptch2$ is expressed in several

Shh-responsive tissues and its upregulation in multiple tissues suggested an elevated Shh pathway activity[57]. We, therefore, looked at the expression pattern of $shha$ in wild-type embryos and detected overlapping expression domains with $ptch2$ in the brain, the floor plate of the neural tube, the pharyngeal arches, and the pectoral fin buds. In $ctcf^{-/-}$ embryos, $shha$ expression domains were differently affected, with reduced expression in the branchial arches and similar wild-type expression in the brain and the floor plate, which contrasts with the elevated expression of its receptor, Ptch2, in those domains (Fig. 6b). Altogether, these results suggest that increased $ptch2$ expression is not due to increased Shh signaling and that CTCF loss disrupts developmental gene regulatory circuits likely due to the impairment of chromatin structure and enhancer–promoter interactions.

Similar changes in the chromatin interactions of RLs and gene expression patterns were observed at the HoxD cluster. Viewpoints from the promoters of $hoxd4a$ and $hoxd13a$ showed reduced interactions within their RLs in $ctcf^{-/-}$ embryos, especially long-range contacts (Fig. 7a). Although we could not detect miss-regulation of $hoxd4a$ and $hoxd13a$ by RNA-seq, in situ hybridization experiments showed a clear reduction of their expression levels (Fig. 7b). However, other $hox$ genes showed consistent miss-regulation detected by both techniques. At the HoxA cluster, we detected a similar reduction in contacts within the RLs of $hoxa5a$ and $hoxa9a$, and reduced gene expression patterns (Supplementary Fig. 12). Altogether, our data indicate that CTCF is required to establish chromatin contacts of gene promoters with their associated RL and to ensure precise spatiotemporal expression patterns of developmental genes.

**Loss of CTCF increases distances within developmental RLs**. To assess whether the changes in expression patterns are a consequence of the observed decrease in promoter interactions within their RLs, we systematically analyzed the enhancer–promoter distances in $ctcf^{-/-}$ embryos. In fact, we used 4Cin[58] to model the three-dimensional architecture of the $ptch2$ and HoxD loci using additional UMI-4C viewpoints as a proxy for spatial distance (Supplementary Fig. 13a, b). 4Cin generated virtual HiC maps representing spatial distances among genomic regions identifying structures that were reminiscent of TADs in our HiC contact matrices (Fig. 8a–c). At the $ptch2$ locus, the TAD containing $ptch2$ was isolated from neighboring TADs in wild-type embryos. In $ctcf^{-/-}$ embryos though, the globular $ptch2$ TAD was less condensed and more intermingled with the adjacent TADs (Fig. 8d). Differential analysis from individual UMI-4C viewpoints illustrated the decrease of long-range contacts within the $ptch2$ TAD, and a general increase with regions outside of it (Supplementary Fig. 13a). We measured the distances of the $ptch2$ promoter within its respective TAD with all ATAC peaks, representing putative CREs, and found a slight general increase in intra-TAD distances in embryos lacking CTCF, although not statistically significant (Fig. 8e). Instead, the distance distribution was generally broader than in wild type. Importantly, we found a similar effect when considering CTCF-bound sites or ATAC peaks with decreased accessibility separately (Supplementary Fig. 13c). This suggests that the altered distances do not result from the loss of CTCF-mediated looping with the promoter or from decreased TF binding to CREs, but rather occur due to a general decompaction of the locus upon CTCF loss.

Using the same approach, we modeled the HoxD locus and found the well-characterized organization of this region in two TADs (called telomeric and centromeric domains, T-DOM and C-DOM, respectively) separated by the cluster of $hoxd$ genes[59] (Fig. 8f–h, Supplementary Fig. 13d). The wild-type 3D model showed this segregation in two globular structures with the gene

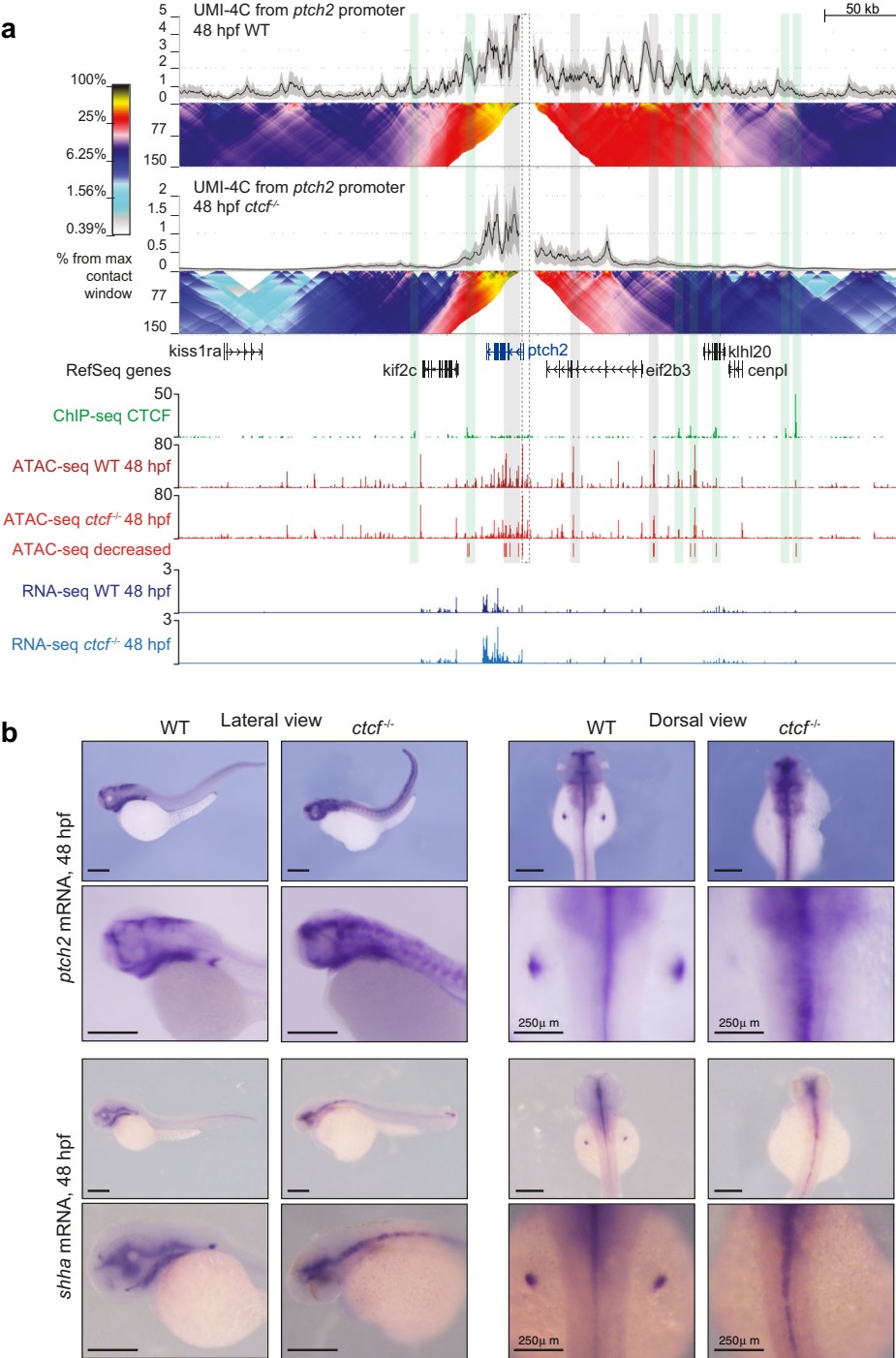

**Fig. 6 CTCF is required to sustain the regulatory landscape and expression pattern of the *ptch2* gene. a** Top, UMI-4C assays in WT and *ctcf*$^{-/-}$ embryos at 48 hpf using the *ptch2* gene promoter as viewpoint. Black lines and gray shadows represent the average normalized UMI counts and their standard deviation, respectively. Domainograms below UMI counts represent contact frequency between pairs of genomic regions. Bottom, tracks with CTCF ChIP-seq, ATAC-seq and RNA-seq at 48 hpf in WT and *ctcf*$^{-/-}$ embryos, as well as decreased ATAC-seq peaks in *ctcf*$^{-/-}$ embryos. A dotted-line square represents the restriction fragment containing the *ptch2* gene promoter that is used as a viewpoint; green shadows highlight CTCF sites and gray shadows highlight downregulated ATAC-peaks without CTCF binding. Upregulated genes are shown in blue. **b** Whole-mount in situ hybridizations of the *ptch2* and *shha* genes in WT and *ctcf*$^{-/-}$ embryos at 48 hpf. Left, lateral view; right, dorsal view. Scale bars represent 500 μm, unless indicated.

cluster in between, whereby *hoxd4a* and *hoxd9a* were located within the designated C-DOM and *hoxd13a* within the T-DOM. This structure appeared less compact and expanded in *ctcf*$^{-/-}$ embryos (Fig. 8i). Distance measurements from the *hoxd4a*, *hoxd9a*, and *hoxd13a* promoters to ATAC peaks within the respective TADs revealed a statistically significant increase in *ctcf*$^{-/-}$ embryos (Fig. 8j,

Supplementary Fig. 13b). Overall, we observed a general loss of compaction within TADs and consequently, increased distances between potential enhancer–promoter pairs in the absence of CTCF. This loss of compaction is accompanied by an increase of inter-TAD interactions, consistent with observations from our promoter HiChIP analysis. We suggest that the loss of wild-type chromatin contacts

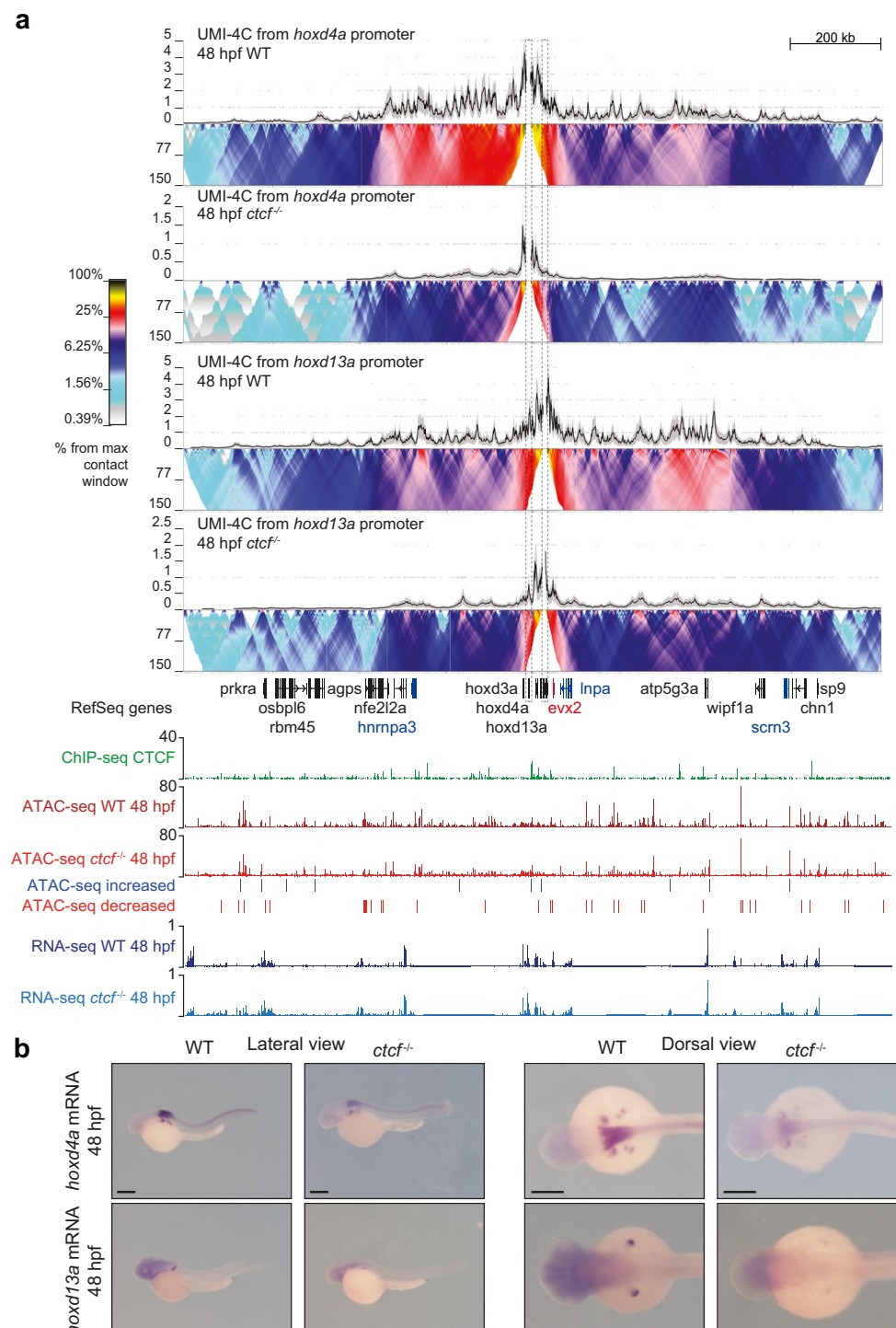

**Fig. 7 CTCF is required to sustain the regulatory landscapes and expression patterns of the *hoxd* genes. a** Top, UMI-4C assays in WT and *ctcf*^−/−^ embryos at 48 hpf using the *hoxd4a* and *hoxd13a* gene promoters as viewpoints. Black lines and gray shadows represent the average normalized UMI counts and their standard deviation, respectively. Domainograms below UMI counts represent contact frequency between pairs of genomic regions. Bottom, tracks with CTCF ChIP-seq, ATAC-seq and RNA-seq at 48 hpf in WT and *ctcf*^−/−^ embryos, as well as increased and decreased ATAC-seq peaks in *ctcf*^−/−^ embryos. Upregulated and downregulated genes are shown in blue and red, respectively. **b** Whole-mount in situ hybridizations of the *hoxd4a* and *hoxd13a* genes in WT and *ctcf*^−/−^ embryos at 48 hpf. Left, lateral view; right, dorsal view. Anterior is to the left and scale bars represent 500 μm.

within RLs and the potential intermingling of interactions across TADs contribute to the observed altered gene expression.

## Discussion

In this work, we have established an in vivo model to study the loss of CTCF in vertebrates, and used multiple epigenomics,

transcriptomic, and chromatin conformation techniques to explore the link between chromatin structure and gene regulation during body plan formation. In the last years, the function of CTCF in chromosome folding has been clearly demonstrated in mammalian in vitro systems, including mouse embryonic stem cells, neural progenitor cells as well as human morula

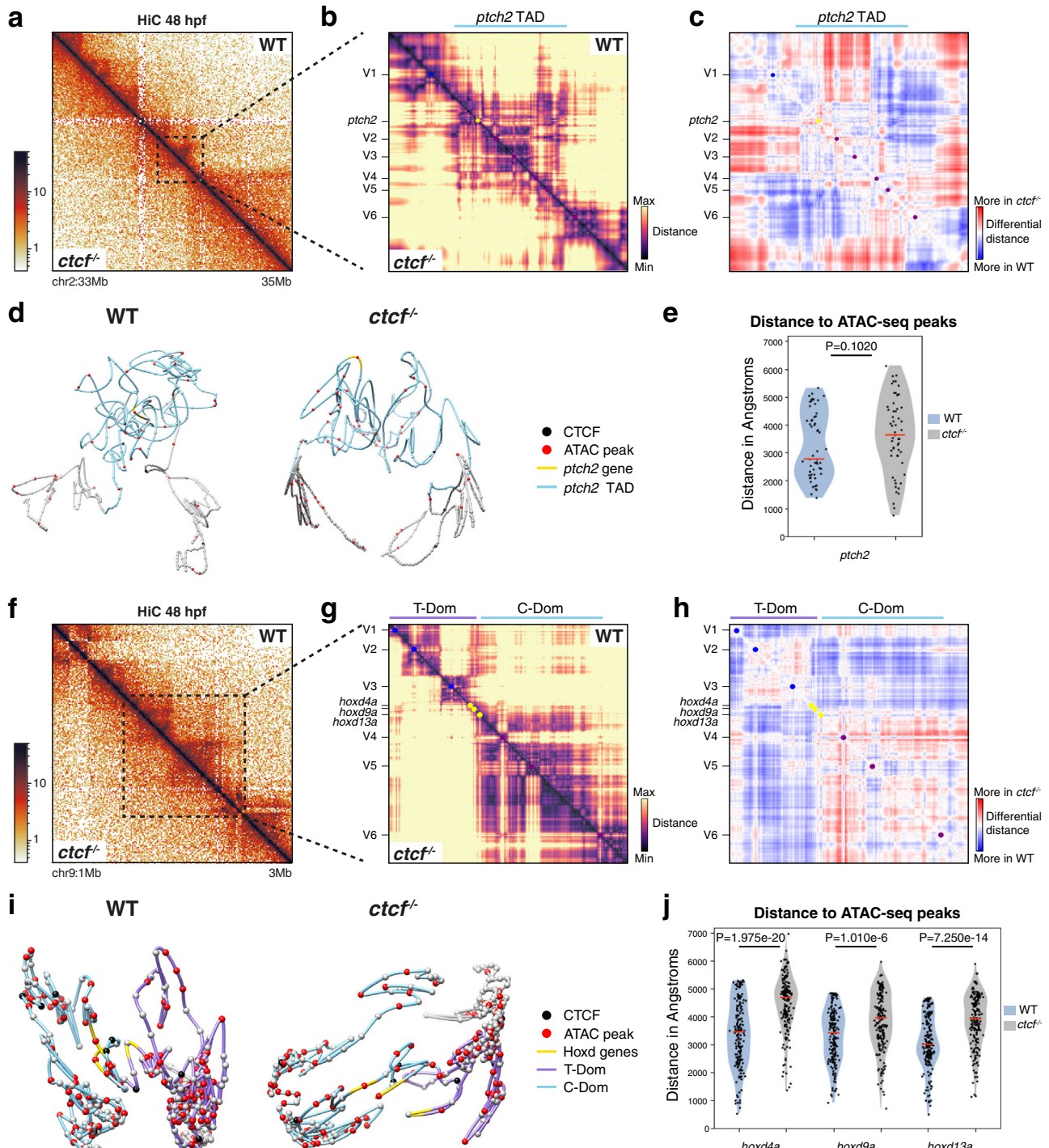

**Fig. 8 3D modeling of developmental loci reveals increased distances within regulatory landscapes. a** HiC normalized contact maps at 10-kb resolution from WT and *ctcf*−/− zebrafish embryos at 48 hpf. A 2-Mb genomic region in chromosome 2 containing the *ptch2* locus is plotted. **b** Virtual HiC matrices of the *ptch2* locus from WT and *ctcf*−/− embryos showing distances based on UMI-4C data at 48 hpf. Viewpoints, including the *ptch2* promoter, are shown on the left. The TAD containing *ptch2* is highlighted in light blue. **c** Differential distances at the *ptch2* locus from (**b**), between WT and *ctcf*−/− embryos. **d** Representative models of the 3D chromatin structure of the *ptch2* locus in WT and *ctcf*−/− embryos. **e** Violin plots showing the distances between ATAC-seq peaks and *ptch2* promoter within *ptch2* TAD. **f** HiC normalized contact maps at 10-kb resolution from WT and *ctcf*−/− zebrafish embryos at 48 hpf. A 2-Mb genomic region in chromosome 9 containing the HoxD cluster is plotted. **g** Virtual HiC matrices of the HoxD locus from WT and *ctcf*−/− embryos showing distances based on UMI-4C data at 48 hpf. Viewpoints, including the *hoxd4a*, *hoxd9a*, and *hoxd13a* promoters, are shown on the left. The two TADs (T- and C-Dom) at the HoxD locus are highlighted in light blue. **h** Differential distances at the HoxD locus from g, between WT and *ctcf*−/− embryos. **i** Representative models of the 3D chromatin structure of the HoxD locus in WT and *ctcf*−/− embryos. **j** Violin plots showing the distances between ATAC-seq peaks in the HoxD locus and *hoxd4a*, *hoxd9a*, and *hoxd13a* promoters. For **e** and **j**, statistical significance was assessed using a two-sided Student's *t* test. ***P < 0.001.

embryos[19,24,25]. These studies showed by different depletion mechanisms that CTCF knockdown severely reduces TAD formation and insulation. Accordingly, we show here that CTCF is also required for chromatin structure in zebrafish embryos (Fig. 1), extending these conclusions to vertebrates and in agreement with a recent report showing that CTCF knockdown in *Xenopus* embryos altered chromatin structure[46].

Despite this well-known function of CTCF, its requirement for the regulation of gene expression has remained controversial. The studies mentioned above showed limited effects of CTCF depletion in gene expression, suggesting that steady-state transcription is mostly resistant to genome-wide alteration of chromatin structure. This contrasts with the observation that CTCF is essential for embryonic development[36], but suggests that CTCF-mediated chromatin structure could be essential for processes in which cells respond to multiple signals and where transcriptional control is highly dynamic. However, the early embryonic lethality of CTCF knockout in animal models has impeded the analysis of CTCF function for transcriptional regulation beyond pluripotency and during the setting up of the animal body plan. Our *ctcf* mutant zebrafish model overcomes this limitation due to the prolonged maternal contribution that lasts until embryo segmentation (Fig. 1). This allows $ctcf^{-/-}$ embryos to develop until stages in which patterning and organogenesis take place. Using this model, we observe the miss-regulation of thousands of genes (Fig. 2), among which many developmental and lineage-specific genes are dynamically regulated during embryonic development. These observations are consistent with recent reports, showing that CTCF is required for the expression of a subset of lineage-specific genes during cell differentiation[25] and for fast transcriptional responses to external stimuli[60].

The expression of developmental genes is characterized by a tight spatiotemporal control by CREs that constitute their RLs. These have been shown to largely coincide with TADs and to be constrained by TAD boundaries[61]. Here we show that the loss of TAD insulation caused by CTCF loss leads to alterations in chromatin contacts of developmental gene promoters (Fig. 3). While reduced contacts involved mostly interactions within the same TAD and often, but not always, coincide with CTCF loops, enhanced contacts frequently span over adjacent TADs and do not contain CTCF. Interestingly, these altered promoter contacts are associated with changes in gene expression, in agreement with recent reports using similar approaches in HeLa cells and in vitro differentiation models[25,56]. Furthermore, the mode of gene miss-expression by differential chromatin contacts is similar to the well-described local reorganization of TADs involving enhancer hijacking and altered gene expression upon genomic structural variations[33]. However, the defects in gene expression that we observe may also be caused by indirect effects due to the function of CTCF in transcriptional regulation. Although our *ctcf* mutant does not allow us to disentangle the multiple CTCF functions, we attempted to quantify the contribution of such indirect effects that are not dependent on its architectural function. We first show that chromatin accessibility at CTCF sites but also at thousands of CREs is compromised in *ctcf* mutants (Fig. 4). Specifically, clusters of CREs within large TADs of developmental genes show highly reduced accessibility, most of them without direct CTCF binding (Fig. 5). Importantly, reduced chromatin accessibility and developmental gene expression arise later than defects in chromatin structure, suggesting that the latter is acutely affected by CTCF loss before changes in chromatin accessibility patterns arise. The decreased accessibility can be explained by the down-regulation of TF genes, which likely induce changes in downstream target gene expression that may contribute to the observed global gene miss-expression. Using differential footprints and network analyses, we exhibit that indirect miss-

regulation by TF genes downstream of CTCF account for less than 50% of all detected DEGs. This suggests other mechanisms, such as altered enhancer–promoter interactions due to lost chromatin structure, to be involved.

To distinguish between TF-induced effects and effects induced by changes in chromatin structure, we analyzed several developmental loci in detail. High-resolution analyses of RLs at the *ptch2*, HoxD, and HoxA loci show that many CREs lose contact with their promoters in the absence of CTCF. These are primarily long-range contacts but without a clear bias for only CTCF sites or for CREs that are differentially regulated (Figs. 6 and 7, Supplementary Fig. 12), suggesting that CTCF-mediated chromatin structure at these loci is required for CRE-promoter contacts independently of TF availability. This is in agreement with our HiChIP data and with previous observations[25,56]. While it is unlikely that CTCF directly mediates those enhancer–promoter interactions, it may favor their establishment by promoting contacts within the involved TADs. It has previously been shown that the loss of CTCF binding sites that contribute to the general TAD structure of a locus is required for CRE–promoter interactions to ensure robust gene expression during development[28,30]. In addition to the reduced contacts within those RLs, we show here that the complex expression patterns of these genes are altered in a tissue-specific manner, showing upregulation and downregulation in different embryonic domains (Fig. 6). This is illustrated by the functionally connected genes *ptch2* and *shha*. While under normal conditions the expression of these genes is co-dependent, in $ctcf^{-/-}$ mutants they have distinct expression patterns in several embryonic domains, suggesting that the absence of CTCF leads to their functional disconnection and gene miss-expression as a consequence of an altered chromatin structure at their respective RLs. Finally, by modeling chromatin interactions and inferring relative distances at the *ptch2* and HoxD loci, we find generally less constrained chromatin interactions within TADs (Fig. 8). Distances between potential enhancers and promoters within these TADs are increased in the absence of CTCF, supporting a role of CTCF-mediated architecture as a facilitator for interactions between promoters and CREs.

In summary, our data demonstrate that CTCF is essential for the correct expression of developmental genes during embryo patterning and organogenesis. Although CTCF influences the expression of a subset of these genes by transcriptional regulation, its function as architectural protein is essential to provide a structural context in which a robust communication between enhancers and their target promoters can take place. This structural context in turn ensures complex spatiotemporal regulation of developmental genes. It has been suggested that TADs may have evolved as conserved scaffolds for developmental gene RLs[62]. Our observations support this view by linking chromatin structure at RLs with gene function.

## Methods

**Animal experimentation.** Wild-type AB/Tübingen zebrafish strains were maintained and bred under standard conditions. All experiments involving animals conform to national and European Community standards for the use of animals in experimentation and were approved by the Ethical Committees from the University Pablo de Olavide, CSIC, and the Andalusian government.

**CRISPR–Cas9 genome editing.** CRISPR target sites to mutate the *ctcf* gene were identified using the CRISPRscan online tool[63]. Two single guide RNAs (sgRNAs) targeting the exons 4 and 5 of the *ctcf* gene were used with the following target sequences: 5′-GGA GTT ACA CTT GCC CAC GC-3′ and 5′-GGC ATG GCC TTT GTC ACC AG-3′. The template DNA for sgRNA transcription was generated by polymerase chain reaction (PCR) using CTCFexon4, CTCFexon5, and sgRNA_universal primers (Supplementary Table 1) and Phusion DNA polymerase (Thermo Fisher Scientific). sgRNAs were in vitro transcribed using the HiScribe T7 Quick High Yield RNA synthesis kit (NEB) using 75 ng of template, treated with

DNase I (NEB), and purified using the RNA Clean and Concentrator kit (Zymo Research).

One-cell stage zebrafish embryos were injected with 2–3 nl of a solution containing 140 ng/µl of Cas9 mRNA and 25 ng/µl of each sgRNA. The CRISPR–Cas9 approach generated a deletion of 260 bp encompassing exons 4 and 5 and resulting in a premature STOP codon in exon 5. The predicted truncated protein had 343 amino acids instead of 798, lacking ten and a half of the eleven zinc finger domains of the CTCF protein. For genotyping, genomic DNA was obtained by incubating the samples (whole embryos or adult caudal fin fragments) in TE buffer supplemented with 5% Chelex-100 (BioRad) and 10 µg/ml Proteinase K (Roche) for 1 h (embryos) or 4 h (fins) at 55 °C and 10 min at 95 °C, and then stored at 4 °C. One microliter of the supernatant was used as a template for standard 20 µl PCR reactions using CTCFpF and CTCFpR primers (Supplementary Table 1), resulting in 842- or 582-bp amplicons for wild type or mutant alleles, respectively. The mutant allele was stably maintained in heterozygosis with no apparent phenotypes, but homozygous mutants are embryonic lethal (<3 days).

**Western blot and protein quantification**. Protein extracts were prepared by resuspending embryos at 48 hpf in loading buffer (100 mM Tris·HCl pH 6.8, 4% sodium dodecyl sulfate (SDS), 0.005% Bromophenol Blue, 20% Glycerol, 205 mM DTT), and then samples were heated at 95 °C for 5 min (for complete protein denaturation).

For western blot, proteins were separated by sodium dodecyl sulfate-polyacrylamide gel electrophoresis, and gels were transferred to nitrocellulose membranes and blocked for 1 h with incubation in TBST (20 mM Tris/HCl, pH 7.4, 150 mM NaCl, and 0.1% [vol/vol] Tween 20) containing 5% (w/vol) skimmed dried milk. Membranes were then incubated for 18 h at 6 °C with the primary CTCF antibody at 1:2000 dilution (PA5-88115, ThermoFisher Scientific) in the same buffer, washed with TBST, and incubated for 1 h at room temperature with the secondary Goat Anti-Rabbit IgG antibody at 1:10,000 dilution (StarBright Blue 520, Bio-Rad, Cat. # 12005870). Membranes were washed with TBST. Dried membranes were imaged, and the signal was quantified using a ChemiDoc MP Image System (Bio-Rad). Stain Free technology was used to record the total protein load for each lane on the transfer membrane and normalization of protein levels was performed as previously described[64], and $ctcf^{+/-}$ and $ctcf^{/}$ lanes were normalized against the WT lane using the software Image Lab 6.0.1 (Bio-Rad). Target protein band volumes of CTCF were corrected using the corresponding normalization factors of total protein load in each lane. Relative CTCF protein levels were calculated from three independent experiments.

**Whole-mount embryo immunofluorescence and protein quantification**. Embryos collected from in-crossed $ctcf^{+/-}$ adult fish were dechorionated and fixed at different developmental stages (1K-cells, 30% epiboly, 80% epiboly, 18 somites stage, 24 hpf). Fixed embryos were washed with PBS-Tween 0.2% (PBT), treated with cold acetone at −20 °C for 20 min, then incubated with freshly prepared blocking solution (2% normal goat serum and 2 mg/mL bovine serum albumin (BSA) in PBT 0.2%) at room temperature for 2 h. A primary antibody specific for zebrafish CTCF[40] was diluted 1:50 in blocking solution and embryos were incubated overnight at 4 °C. Embryos were subsequently washed with PBT and incubated overnight at 4 °C in the dark with the Alexa Fluor TM 555 Goat anti-rabbit antibody (Invitrogen #A32727), diluted 1:500. Finally, embryos were washed with PBT and incubated overnight at 4 °C with DAPI (Sigma) diluted 1:5000 and Alexa Fluor TM 488 phalloidin (Invitrogen #A12379) diluted 1:100 in PBT 0.2%.

For imaging, embryos were embedded in 1% low-melting-point agarose, transferred to glass-bottom culture dishes (MatTek Corporation), and manually oriented. Only embryos that were mounted with the vegetal–animal axis completely parallel to the cover glass were used for analysis. Confocal laser scanning microscopy was performed using an LSM 880 microscope (Zeiss). Images were processed using Fiji. The signal intensity of CTCF and DAPI staining was quantified in the whole embryo for all stages over a depth of 140 µm using the measure tool in Fiji. In addition, anterior (equivalent to neural developing tissues) and posterior (equivalent to notochord and tail tissues) regions were also quantified separately in embryos at 18 somite stage and 24 hpf. Values for CTCF intensity were normalized by DAPI intensity to avoid changes in CTCF signals related to differences in nuclear density.

After imaging, embryos were genotyped by PCR to identify $ctcf$-related genotypes. To analyze whether experimental groups were significantly different, two-sided Student's $t$ tests ($\alpha = 0.05$) were performed.

**Whole-mount embryo in situ hybridization**. Antisense RNA probes were prepared from cDNA using digoxigenin (Boehringer Mannheim) as a label and the primers listed in Supplementary Table 1, except those for $shha$ and $hoxd13a$ that were previously described[65]. Zebrafish embryos were prepared, hybridized, and stained using standard protocols[66]. Embryos at 48 hpf stage were fixed in 4% paraformaldehyde overnight, dehydrated in methanol, and stored at −20 °C. All solutions and reagents used were RNAse-free. The embryos were hydrated using decreasing amounts of methanol and finally in PBS-0.1% Tween. Then, they were treated with 10 µg/ml proteinase K for 10 min at room temperature and gently washed with PBS-0.1% Tween. In the pre-hybridization step, embryos were kept at

70 °C in the hybridization buffer for at least 1 h. Then, the probe was diluted to 2 ng/µl in hybridization buffer and incubated overnight at 70 °C while moving. Pre-heated buffers with decreasing amounts of hybridization buffer (75%, 50%, 25%, and 0%) in 2× SSC solution were used to wash embryos for 10 min, plus a 30 min wash at 70 °C with 0.05× SSC. Then, they were incubated with Blocking Buffer (PBS-0.1% Tween, 2% normal goat serum, 2 mg/ml bovine serum albumin [BSA]) for 1 h, and with an anti-digoxigenin antibody (1:5000 in Blocking Buffer) for at least 2 h at room temperature. After this, embryos were washed six times with PBS-0.1% Tween at room temperature and then overnight at 4 °C. The next day, embryos were washed once more with PBS-0.1% Tween and three times with fresh AP buffer (100 mM Tris-HCl pH 9.5, 50 mM MgCl2, 100 mM NaCl, 0.1% Tween), followed by signal development with NBT/BCIP solution (225 µg/ml NBT, 175 µg/ml BCIP) in multi-well plates in the dark. Signal development was stopped by washing with PBS-0.1% Tween and fixing with 4% paraformaldehyde. Imaging of the in situ hybridization signal was performed in the MZ-12 dissecting scope (Leica).

**RNA-seq**. For total RNA extraction, wild-type and $ctcf^{-/-}$ single embryos at 24 or 48 hpf were collected, manually dechorionated, and suspended in TRIsure (Bioline) with chloroform. DNA was used for genotyping and single wild-type and $ctcf^{-/-}$ individuals were selected for RNA-seq experiments. Precipitated RNA was then treated with TURBO DNA-free kit (Invitrogen). Two biological replicates were used for each analyzed genotype and stage.

Illumina libraries were constructed and sequenced in a BGISEQ-500 single-end lane producing around 50 million (M) of 50-bp reads. Reads were aligned to the GRCz10 (danRer10) zebrafish genome assembly using STAR 2.5.3a[67] and counted using the htseq-count tool from the HTSeq 0.8.0 toolkit[68]. Differential gene expression analysis was performed using the DESeq2 1.18.1 package in R 3.4.3[69], setting a corrected $P$ value < 0.01 as the cutoff for statistical significance of the differential expression. Enrichment of GO Biological Process terms was calculated using David 6.8[70], with a false discovery rate (FDR)-corrected $P$ value < 0.05 as the statistical cutoff.

**ATAC-seq**. ATAC-seq assays were performed using standard protocols[71,72], with minor modifications. Briefly, single WT or $ctcf^{-/-}$ mutant embryos at 24 or 48 hpf coming from $ctcf^{+/-}$ crosses were manually dechorionated. The yolk was dissolved with Ginzburg Ring Finger (55 mM NaCl, 1.8 mM KCl, 1.15 mM NaHCO3) by pipetting and shaking 5 min at 1100 rpm. Deyolked embryos were collected by centrifugation for 5 min at 500$g$ 4 °C. The supernatant was removed and embryos were washed with PBS. Then, embryos were lysed in 50 µl of Lysis Buffer (10 mM Tris-HCl pH 7.4, 10 mM NaCl, 3 mM MgCl2, 0.1% NP-40, 1× Roche Complete protease inhibitors cocktail) by pipetting up and down. The whole-cell lysate was used for TAGmentation, which was centrifuged for 10 min at 500 g 4 °C and resuspended in 50 µl of the Transposition Reaction, containing 1.25 µl of Tn5 enzyme and TAGmentation Buffer (10 mM Tris-HCl pH 8.0, 5 mM MgCl2, 10% w/v dimethylformamide), and incubated for 30 min at 37 °C. Immediately after TAGmentation, DNA was purified using the Minelute PCR Purification Kit (Qiagen) and eluted in 20 µl. Before library amplification, purified DNA was used to genotype 24-hpf embryos (see above) and wild-type or $ctcf^{-/-}$ mutants were selected for deep sequencing. Libraries were generated by PCR amplification using NEBNext High-Fidelity 2× PCR Master Mix (NEB). The resulting libraries were multiplexed and sequenced in a HiSeq 4000 pair-end lane producing 100 M of 49-bp pair-end reads per sample.

**ChIPmentation**. ChIP-seq of CTCF was performed by ChIPmentation, which incorporates Tn5-mediated TAGmentation of immunoprecipitated DNA, as previously described[73,74]. Briefly, 100 zebrafish embryos at 24 hpf were dechorionated with 300 µg/ml pronase, fixed for 10 min in 1% paraformaldehyde (in 200 mM phosphate buffer) at room temperature, quenched for 5 min with 0.125 M glycine, washed in PBS, and frozen at −80 °C. Fixed embryos were homogenized in 2 ml cell lysis buffer (10 mM Tris-HCl pH 7.5, 10 mM NaCl, 0.3% NP-40, 1× Roche Complete protease inhibitors cocktail) with a Dounce Homogenizer on ice and centrifuged 5 min 2300$g$ at 4 °C. Pelleted nuclei were resuspended in 333 µl of nuclear lysis buffer (50 mM Tris-HCl pH 7.5, 10 mM EDTA, 1% SDS, 1× Roche Complete protease inhibitors cocktail), kept 5 min on ice and diluted with 667 µl of ChIP dilution buffer (16.7 mM Tris-HCl pH 7.5, 1.2 mM EDTA, 167 mM NaCl, 0.01% SDS, 1.1% Triton-X100). Then, chromatin was sonicated in a Covaris M220 sonicator (duty cycle 10%, PIP 75 W, 100 cycles/burst, 10 min) and centrifuged 5 min 18,000$g$ at 4 °C. The recovered supernatant, which contained soluble chromatin, was used for ChIP or frozen at −80 °C after checking the size of the sonicated chromatin. Four 250 µl aliquots of sonicated chromatin were used for 4 independent ChIP experiment, and each aliquot was incubated with 2 µg of anti-CTCF antibody[40] and rotated overnight at 4 °C. The next day, 20 µl of protein G Dynabeads (Invitrogen) per aliquot were washed twice with ChIP dilution buffer and resuspended in 50 µl/aliquot of the same solution. Immunoprecipitated chromatin was then incubated with washed beads for 1 h rotating at 4 °C and washed twice sequentially with wash buffer 1 (20 mM Tris-HCl pH 7.5, 2 mM EDTA, 150 mM NaCl, 1% SDS, 1% Triton-X100), wash buffer 2 (20 mM Tris-HCl pH 7.5, 2 mM EDTA, 500 mM NaCl, 0.1% SDS, 1% Triton-X100), wash buffer 3

(10 mM Tris-HCl pH 7.5, 1 mM EDTA, 250 mM LiCl, 1% NP-40, 1% Na-deoxycholate) and 10 mM Tris-HCl pH 8.0, using a cold magnet (Invitrogen). Then, beads were resuspended in 25 μl of TAGmentation reaction mix (10 mM Tris-HCl pH 8.0, 5 mM MgCl2, 10% w/v dimethylformamide), added 1 μl of Tn5 enzyme and incubated 1 min at 37 °C. TAGmentation reaction was put in the cold magnet and the supernatant was discarded. Beads were washed twice again with wash buffer 1 and 1× TE and eluted twice for 15 min in 100 μl of elution buffer (50 mM NaHCO3 pH 8.8, 1% SDS). The 200 μl of eluted chromatin per aliquot were then decrosslinked by adding 10 μl of 4 M NaCl and 1 μl of 10 mg/ml proteinase K and incubating at 65 °C for 6 h. DNA was purified using Minelute PCR Purification Kit (Qiagen), pooling all aliquots in a single column, and eluted in 20 μl. Library preparation was performed as previously described for ATAC-seq (see above). Libraries were multiplexed and sequenced in a HiSeq 4000 pair-end lane producing around 20 M of 49-bp paired-end reads per sample.

**ChIPmentation and ATAC-seq data analyses.** ChIPmentation and ATAC-seq reads were aligned to the GRCz10 (danRer10) zebrafish genome assembly using Bowtie2 2.3.5[75] and those pairs separated by more than 2 kb were removed. For ATAC-seq, the Tn5 cutting site was determined as the position −4 (minus strand) or +5 (plus strand) from each read start, and this position was extended 5 bp in both directions. Conversion of SAM alignment files to BAM was performed using Samtools 1.9[76]. Conversion of BAM to BED files, and peak analyses, such as overlaps or merges, were carried out using the Bedtools 2.29.2 suite[77]. Conversion of BED to BigWig files was performed using the genomecov tool from Bedtools and the wigToBigWig utility from UCSC[78]. For ATAC-seq, peaks were called using MACS2 2.1.1.20160309 algorithm[79] with an FDR < 0.05 for each replicate and merged in a single pool of peaks that was used to calculate differentially accessible sites with DESeq2 1.18.1 package in R 3.4.3[69], setting a corrected P value < 0.01 as the cutoff for statistical significance of the differential accessibility. For ChIP-mentation, peaks with an FDR < 0.001 were called with MACS2. For visualization purposes, reads were extended 100 bp for ATAC-seq and 300 bp for ChIPmentation. For data comparison, all ATAC-seq experiments used were normalized using reads falling into peaks to counteract differences in background levels between experiments and replicates[73].

Heatmaps and average profiles of ChIPmentation and ATAC-seq data were generated using computeMatrix, plotHeatmap and plotProfile tools from the Deeptools 3.5 toolkit[80]. TF motif enrichment and peak annotation to genomic features were calculated using the scripts FindMotifsGenome.pl and AnnotatePeaks.pl from Homer 4.11 software[81], with standard parameters. For gene assignment to ChIP and ATAC peaks, coordinates were converted to Zv9 (danRer7) genome using the Liftover tool from the UCSC Genome Browser[78] and assigned to genes using the GREAT 3.0.0 tool[82], with the basal plus extension association rule with standard parameters (5 kb upstream, 1 kb downstream, 1 Mb maximum extension). Peak clustering was calculated using the mergeBed tool from Bedtools[77], considering as clustered those peaks located less than 30 kb from each other.

For footprinting analyses, we used TOBIAS 0.12.9[54]. First, we performed bias correction using ATACorrect and calculated footprint scores with ScoreBigwig, both with standard parameters. Then, we used BINDetect to determine the differential TF binding for all vertebrate motifs in the JASPAR database[83]. We considered as differentially bound those motifs with a linear fold-change ≥15% between WT and ctcf−/− embryos, and whose TF genes show curated expression in 48-hpf zebrafish embryos according to the ZFIN database. Aggregated ATAC-seq signal at footprints was visualized using PlotAggregate. Finally, the pool of TF binding sites from differentially bound TFs was used to build a TF binding network starting in CTCF and reaching three levels. This network was visualized using Cytoscape v3.8.2[84].

**HiC.** HiC library preparation was performed as previously described[10] with minor modifications. Experiments were performed for at least two biological replicates in wild-type and ctcf−/− mutant embryos at 24 and 48 hpf, using one to three million cells as input material. At 24 hpf, 150 individual embryos from ctcf+/− crossings were dissociated and 5/6 of dissociated cells were used for fixation with 1% par-aformaldehyde as described below and then stored at −80 °C. The remaining 1/6 of dissociated cells was used for genotyping. Wild type or ctcf+/− mutants were selected and pooled for chromatin digestion as described below.

*Embryo fixation and nuclei extraction.* Pools of 50 zebrafish embryos were dechorionated with 300 μg/ml pronase, followed by fixation for 10 min in 1% paraformaldehyde (in 200 mM phosphate buffer) at room temperature. The reaction was quenched by adding glycine to a final concentration of 0.125 M and incubating at room temperature for 5 min. Embryos were washed on ice twice with 1× PBS and either snap-frozen in liquid nitrogen or processed for nuclei extraction. For nuclei extraction, fixed embryos were homogenized in 2–5 ml freshly prepared lysis buffer (50 mM Tris pH7.5; 150 mM NaCl; 5 mM EDTA; 0.5% NP-40; 1.15% Triton X-100; 1× Roche Complete protease inhibitors) with a Dounce Homo-genizer on ice. Nuclei were pelleted by centrifugation for 5 min, 750 g at 4 °C, and washed with 1× PBS. Pelleted nuclei were either snap-frozen in liquid nitrogen or further processed.

*Chromatin digestion.* Nuclei pellets were resuspended in 100 μl 0.5% SDS and incubated for 10 min at 62 °C, without shaking. Totally, 292 μl water and 50 μl 10% Triton X-100 was added to each sample, mixed, and incubated for 15 min at 37 °C to quench the remaining SDS. A 50 μl of 10× restriction enzyme buffer and a total of 400 units of DpnII (NEB, R0543) were added to the sample, mixed, and incubated overnight at 37 °C with 900 rpm shaking.

*Biotin fill-in and proximity ligation.* Restriction enzyme was heat-inactivated. Nuclei were pelleted at 600 g for 10 min at 4 °C and resuspended in 445 μl 1× ice-cold NEB buffer 2. For biotin fill-in reaction, 5 μl of 10× NEB buffer 2, 1.5 μl 10 mM (each) dNTP-dATP-mix, 37.5 μl of 0.4 mM biotin-14-dATP, and 10 μl of 5 U/μl Klenow (NEB, M0210L) were added and mixed by pipetting. Samples were incubated at 25 °C for 4 h and 800 rpm shaking. To ligate restriction fragment ends, 500 μl of 2× ligation mix (100 μl of 10× ligation buffer (NEB), 100 μl of 10% Triton-X-100, 10 μl of 10 mg/ml BSA, 6.5 μl of T4 DNA ligase (NEB, M0202L), 283.5 μl water) were added to each sample and incubated overnight at 16 °C and 800 rpm shaking.

*Cross-link reversal and DNA purification.* Nuclei were pelleted by centrifugation for 10 min, 600 g at 4 °C, and sample volume was reduced to a total of 200 μl. Totally, 230 μl of 10 mM Tris HCL pH7.5, 20 μl of Proteinase K (10 mg/ml) and 50 μl of 10% SDS were added, mixed by pipetting, and incubated 30 min at 55 °C. Subsequently, 40 μl of 4 M NaCl were added and samples were incubated overnight at 65 °C with 700 rpm shaking. Next, 5 μl of RNAse A (10 mg/ml) were added, followed by incubation at 37 °C for 30 min at 700 rpm. 20 μl Proteinase K (10 mg/ml) were added to the sample and incubated at 55 °C for 1–2 h at 700 rpm. DNA was purified by phenol-chloroform extraction. Following DNA precipitation, the dried DNA pellet was reconstituted in 100 μl 10 mM Tris-HCl pH 7.5.

*Removing biotin from un-ligated fragments and DNA shearing.* A 5–7 μg of HiC library in a total volume of 100 μl (1× NEB buffer 2.1, 0.025 mM dNTPs, 0.12 U/μl T4 DNA polymerase (NEB, M0203) was incubated at 20 °C for 4 h to remove biotin from unligated ends. The reaction was stopped by adding EDTA to a final concentration of 10 mM and heat inactivated for 20 min at 75 °C. DNA was sheared, using Covaris M220 sonicator with the following setup: 130 μl sample volume, Peak Incident Power (W): 50, Duty Factor: 20%, Cycles per Burst: 200, Treatment Time (s): 65, cooling at 7 °C. Samples were subsequently size selected for fragments between 150 and 600 bp using AMPure XP beads (Agencourt, A63881) as follows: 0.575× volume of AMPure beads were added to the sample, mixed by pipetting, and incubated for 10 min at room temperature. Beads were separated on a magnet, and the clear supernatant was transferred to a fresh tube. 0.395× volume of fresh AMPure beads were added to the supernatant, mixed, and incubated for 10 min at room temperature. Beads were separated on a magnet, and the clear supernatant was discarded. Beads were washed twice with 70% EtOH, air-dried for 5 min and DNA was eluted in 300 μl water.

*Biotin pull-down.* Biotin-labeled DNA was bound to Dynabeads My One C1 Streptavidin beads, using 5 μl of beads per 1 μg DNA and following manufacturer's instructions. Beads were washed twice with 1× tween-washing buffer (5 mM Tris HCl pH 7.5, 0.5 mM EDTA, 1 M NaCl, 0.05% Tween 20) and finally resuspended in 1× sample volume 2× binding buffer (10 mM Tris HCl pH 7.5, 1 mM EDTA, 2 M NaCl). Beads were mixed with the DNA sample and incubated for 20 min at room temperature while rotating. Beads were separated on a magnet, twice washed with 1× tween-washing buffer at 55 °C and 700 rpm shaking for 2 min. Reclaimed beads were resuspended in 50 μl water.

*Sequencing library preparation.* To repair DNA ends, DNA-bound beads were incubated in 100 μl end-repair mix containing 1× T4 Ligase Buffer (NEB), 0.5 mM dNTP mix, 0.5 U/μl T4 Polynucleotide Kinase (NEB, M0201), 0.12 U/μl T4 DNA Polymerase (NEB, M0203), and 0.05 U/μl Klenow (NEB, M0210). Samples were incubated for 30 min at 20 °C. Beads were separated on a magnet, twice washed with 1× tween-washing buffer at 55 °C and 700 rpm shaking for 2 min. Reclaimed beads were resuspended in 50 μl water. Next, dA-tail was added by incubating DNA-bound beads in 100 μl A-tailing mix, containing 1× NEB buffer, 0.5 μM dATP, and 0.25 U/μl Klenow, exo- (NEB, M0212). Samples were incubated for 30 min at 37 °C. Beads were separated on a magnet, twice washed with 1× tween-washing buffer at 55 °C and 700 rpm shaking for 2 min. Reclaimed beads were resuspended in 20 μl water. Subsequently, samples were indexed by ligating TruSeq Illumina adapters by incubating DNA-bound beads in 50 μl adapter ligation mix, containing 1× T4 Ligation buffer, 5% PEG-4000, 0.3 U/μl T4 DNA Ligase (ThermoFisher, EL0011), 1.5 μl TruSeq index adapter. The reaction was incubated at 22 °C for 2 h with occasional mixing. Beads were separated on a magnet, twice washed with 1× tween-washing buffer at 55 °C and 700 rpm shaking for 2 min. Reclaimed beads were resuspended in 50 μl water. The final library for paired-end sequencing was prepared using NEBNext High-Fidelity 2× PCR Master Mix (NEB). PCR reaction: 50 μl reaction, containing 1× NEBNext High-Fidelity PCR Master Mix, 0.3 μM TruSeq Primer 1.0 (P5) and TruSeq Primer 2.0 (P7), 3 μl DNA-bound beads. PCR cycler setup: (1) 98 °C for 60 s, (2) 98 °C for 10 s, (3) 65 °C for 30 s, (4) 72 °C for 30 s, (5) Go to step 2 for up to 10 cycles, and (6) 72 °C for 5 min. The optimal cycle number was determined for each sample by analyzing a

5 µl aliquot on an agarose gel after 4, 6, 8, 10, and 12 cycles. For each sample, at least eight independent PCR reactions were performed to maintain initial library complexity and then pooled for AMPure beads purification. 1.2× volume of AMPure beads was added to the sample, mixed by pipetting, and incubated for 10 min at room temperature. Beads were separated on a magnet, and the clear supernatant was discarded. Beads were washed twice with 70% EtOH and air-dried for 5 min. DNA was eluted in 50 µl water. Libraries were multiplexed and sequenced using DNBseq technology to produce 50 bp paired-end reads and approximately 400 million raw sequencing read pairs for each genotype.

### HiC data analyses

*Mapping, filtering, normalization, and visualization.* HiC paired-end reads were mapped to the zebrafish genome assembly GRCz10 (danRer10) using BWA[85]. Reads from biological replicates were pooled before mapping. Then, ligation events (HiC pairs) were detected and sorted, and PCR duplicates were removed, using the pair tools package (https://github.com/mirnylab/pairtools). Unligated and self-ligated events (dangling and extra-dangling ends, respectively) were filtered out by removing contacts mapping to the same or adjacent restriction fragments. The resulting filtered pairs file was converted to a tsv file that was used as input for Juicer Tools 1.13.02 Pre[86], which generated multiresolution hic files. These analyses were performed using custom scripts (https://gitlab.com/rdacemel/hic_ctcf-null): the hic_pipe.py script was first used to generate tsv files with the filtered pairs, and the filt2hic.sh script was then used to generate Juicer hic files. HiC matrices at 10 and 500 kb resolution, normalized with the Knight–Ruiz (KR) method[87], were extracted for downstream analysis using the FAN-C 0.9.14 toolkit[88]. Visualization of normalized HiC matrices and other values described below, such as insulation scores, TAD boundaries, aggregate TAD, and loop analysis, Pearson's correlation matrices and eigenvectors, were calculated and visualized using FAN-C.

*TADs, chromatin loops, and compartmentalization.* TAD boundaries were called using the insulation score method[45]. Insulation scores were calculated for 10-kb binned HiC matrices using FAN-C[88]. Briefly, the average number of interactions of each bin was calculated in 500-kb square sliding windows ($50 \times 50$ bins); then, these values were normalized as the $\log_2$ ratio of each bin's value and the mean of all bins to obtain the insulation score for each bin; next, minima along the insulation score vector were calculated using a delta vector of ±100 Kb (±10 bins) around the central bin; finally, boundaries with scores lower than 0.5 were filtered out. The genomic regions located between adjacent boundaries were considered as TADs.

For the determination of A and B compartments, 100-kb binned HiC matrices were used. Pearson's correlation matrices were calculated as previously described[89], using FAN-C[88]. A and B compartments were determined using the values of the second eigenvector since the first eigenvector corresponded to chromosome arms in our experiments. The A compartment was assigned as the one with the highest enrichment in H3K27ac, H3K27me3, H3K4me3, RNA-seq, and ATAC-seq signals. A/B domains were defined as consecutive regions with the same eigenvector sign. Enrichment profiles were calculated by ranking A and B compartments in fifty percentiles according to their eigenvalues and plotting their average observed/expected contact values. AB strength was calculated as the natural logarithm of AA * BB/AB[2], where AA, BB, and AB represent interactions among A or B domains. Interactions among A and B compartments were quantified separately for biological replicates at 500-kb resolution.

Chromatin loops were called using Juicer Tools 1.13.02 Hiccups[10], with standard parameters. Briefly, the multiresolution hic file was used as input for the CPU version of HICCUPS, which runs using 5, 10, and 25-kb resolution KR-normalized matrices. The maximum permitted FDR value was 0.1 for the three resolutions; the peak widths were 4, 2, and 1 bin for 5, 10, and 25-kb resolutions, respectively; and the window widths to define the local neighborhoods used as background were 7, 5, and 3 bins, respectively. The thresholds for merging loop lists from different resolutions were the following: maximum sum of FDR values of 0.02 for the horizontal, vertical, donut and lower-left neighborhoods; minimum enrichment of 1.5 for the horizontal and vertical neighborhoods; minimum enrichment of 1.75 for the donut and bottom-left neighborhoods; minimum enrichment of 2 for either the donut or the bottom-left neighborhoods. The distances used to merge the nearby pixels to a centroid were 20, 20, and 50-kb for 5, 10, and 25-kb resolutions, respectively. CTCF-bound and chromatin loops were considered when at least one of the loop anchors overlapped with a CTCF ChIP-seq peak.

### HiChIP

HiChIP assays were performed as previously described[52], with modifications. Briefly, 150 WT or $ctcf^{-/-}$ zebrafish embryos at 48 hpf stage were dechorionated with 300 µg/ml pronase and transferred to 1 ml of Ginzburg fish ringer buffer (55 mM NaCl, 1.8 mM KCl, 1.25 mM NaHCO3). Yolks were disrupted by pipetting and shaking for 5 min at 1100 rpm. Embryos were then spinned-down and fixed as indicated above for ChIPmentation. Fixed embryos were homogenized in 5 ml cell lysis buffer (see above) with a Dounce Homogenizer on ice. Complete cell lysis generating nuclei was checked at the microscope with methylgreen-pyronin staining. Nuclei were then centrifuged 5 min at 600 g at 4 °C and in situ contact generation was performed as described[52], with modifications. For chromatin digestion, 8 µl of 50 U/µl DpnII restriction enzyme were used. Ligation was incubated overnight at 16 °C shaking at 900 rpm. Both digestion and ligation efficiencies were monitored by taking control aliquots that were de-crosslinked,

phenol–chloroform purified, and loaded in an 0.7% agarose gel. The controls consisted of 5-µl aliquots before and after digestion (undigested and digested controls), a 10-µl aliquot before end repair that was ligated with overhang ends (3 C control), and a 25-µl aliquot after ligation (ligation control).

After ligation, nuclei were pelleted at 2500g for 5 min at RT, resuspended in 495 µl of nuclear lysis buffer (see above), and kept 5 min on ice to lysate nuclei. Then, 1980 µl of ChIP dilution buffer (see above) was added and the sample was split into three 1-ml aliquots that were sonicated in a Covaris M220 sonicator (duty cycle 10%, PIP 75 W, 100 cycles/burst, 5 min). Then, sonicated chromatin was centrifuged for 15 min at 16,000g at 4 °C and the supernatant was transferred to a new tube. Sonication efficiency was checked using a 20-µl aliquot that was RNase A-treated, de-crosslinked, phenol-chloroform purified, and loaded in a 0.7% agarose gel. After this, chromatin was pre-cleared with Dynabeads Protein G (Invitrogen) rotating for 1 h at 4 °C, recovered to new tubes using a magnet and incubated overnight rotating at 4 °C with 6.7 µg (20 µg total) of anti-H3K4me3 (Abcam ab8580) antibody per sample. Immunoprecipitated chromatin was then washed and eluted from beads as described[52]. Before DNA purification, the three samples were mixed and split in two samples to generate later two independent libraries, increasing the likelihood of library amplification over primer artifact amplification.

Biotin capture with Streptavidin C-1 beads (Invitrogen) and TAGmentation was performed as described[52]. For library preparation, samples were put in a magnet, supernatant discarded, and beads resuspended in a 50-µl PCR mix containing 1× NEBNext High-Fidelity PCR Master Mix (NEB) and 0.5 µM of Nextera Ad1_noMX and Ad2.X primers. PCR was run for 5 cycles and then samples were put in a magnet to separate beads. Then, the cycle number for library preparation was estimated by qPCR taking a 2-µl aliquot from the samples, and the remaining PCR was run for the empirically determined number of cycles. Finally, libraries were recovered from beads using a magnet, pooled together, and purified using DNA Clean and Concentrator columns (Zymo Research), eluting in 20 µl of 10 mM Tris-HCl pH 8.0. Libraries were quantified in a Qubit machine and sequenced using DNBseq technology to generate around 500 M of 50-bp paired-end reads.

### HiChIP data analyses

HiChIP paired-end reads were aligned to GRCz10 (danRer10) zebrafish genome assembly using the TADbit pipeline[90]. Default settings were used to remove duplicate reads, assign reads to DpnII restriction fragments, filter for valid interactions, and generate binned interaction matrices with a 10-kb resolution. Data were visualized using the WashU Epigenome Browser[91]. Since HiC normalization methods are not suitable for HiChIP data given the inherent scarcity of HiChIP matrices, we scaled the samples to the same number of valid read pairs.

HiChIP loops for individual biological replicates were calculated using FitHiChIP 9.0[53] with the following parameters: interaction type peak to all, bin size 10 kb, distance threshold between 20 kb and 20 Mb, FDR < 0.01, loose background for contact probability estimation [FitHiChIP(L)], coverage bias regression and merging of redundant loops. For differential analysis of HiChIP loops between WT and $ctcf^{-/-}$ embryos, we considered a stringent set of loops consisting of the merge of those detected in both biological replicates per condition, with a differential FDR threshold of 0.05 and a fold-change threshold of 1.5. To avoid calling loops as differential due to different ChIP-seq coverage, we only considered loops involving H3K4me3 peaks not called as differential by EdgeR (i.e., categories ND-ND, LD-LD, and ND-LD from the FitHiChIP differential analysis output).

### UMI-4C

UMI-4C library preparation was performed as previously described[55] with modifications in 3C library preparation and minor modifications in sequencing library preparation. Experiments were performed in singletons in wild-type and $ctcf^{-/-}$ mutant embryos at 48 hpf, using one to three million cells as input material. Embryo fixation, nuclei extraction, chromatin digestion, biotin fill-in, proximity ligation, cross-link reversal, and DNA purification were performed following the above experimental procedure for HiC. The following procedure was specific for UMI-4C.

*DNA shearing.* A 5–7 µg of purified DNA was sheared with Covaris M220 sonicator with the following setup: 130 µl sample volume, Peak Incident Power (W): 50, Duty Factor: 10%, Cycles per Burst: 200, Treatment Time (s): 70, cooling at 7 °C. Samples were then purified using AMPure XP beads (Agencourt, A63881) as follows: 2.0× volume of AMPure beads was added to the sample, mixed by pipetting, and incubated for 10 min at room temperature. Beads were separated on a magnet, and the clear supernatant was discarded. Beads were washed twice with 70% EtOH, and air-dried for 5 min. DNA was eluted in 300 µl water.

*Biotin pull-down.* Biotin-labeled DNA was bound to Dynabeads My One C1 Streptavidin beads, using 5 µl of beads per 1 µg DNA and following manufacturer's instructions. Beads were washed twice with 1× tween-washing buffer (5 mM Tris HCl pH 7.5, 0.5 mM EDTA, 1 M NaCl, 0.05% Tween 20) and finally resuspended in 1× sample volume 2× binding buffer (10 mM Tris HCl pH 7.5, 1 mM EDTA, 2 M NaCl). Beads were mixed with the DNA sample and incubated for 20 min at room temperature while rotating. Beads were separated on a magnet, twice washed

with 1× tween-washing buffer at 55 °C and 700 rpm shaking for 2 min. Reclaimed beads were resuspended in 50 μl water.

*Sequencing library preparation.* A 500 ng of DNA attached to beads were end-repaired by incubating in 100 μl end-repair mix (1× T4 Ligase Buffer (NEB), 0.5 mM dNTP mix, 0.12 U/μl T4 DNA Polymerase (NEB, M0203) and 0.05 U/μl Klenow (NEB, M0210)) for 30 min at 20 °C. Beads were separated on a magnet, twice washed with 1× tween-washing buffer at 55 °C and 700 rpm shaking for 2 min. Reclaimed beads were resuspended in 50 μl water. Next, DNA-bound beads were incubated for 30 min at 37 °C in 100 μl A-tailing mix (1× NEB buffer, 0.5 μM dATP, and 0.25 U/μl Klenow, exo- (NEB, M0212)). The enzyme was heat-inactivated at 75 °C for 20 min. For 5′ dephosphorylation of DNA ends, 2 μl of Alkaline Phosphatase, Calf Intestinal (NEB, M0290) were added, and samples were incubated at 37 °C for 1 h and with occasionally mixing. Beads were separated on a magnet, twice washed with 1× tween-washing buffer at 55 °C and 700 rpm shaking for 2 min. Reclaimed beads were resuspended in 20 μl water. Next, samples were indexed by ligating TruSeq Illumina adapters by incubating DNA-bound beads in 50 μl adapter ligation mix (1× T4 Ligation buffer, 5% PEG-4000, 0.3 U/μl T4 DNA Ligase (ThermoFisher, EL0011), 1.5 μl TruSeq index adapter). The reaction was incubated at 22 °C for 2 h with occasional mixing. The sample volume was increased with water to a total of 100 μl and incubated at 96 °C for 5 min to denature DNA and remove non-ligated strand from adapter. The sample was placed on ice and beads were separated on a magnet, twice washed with 1× tween-washing buffer at 55 °C and 700 rpm shaking for 2 min. Reclaimed beads were resuspended in 20 μl water. The final library for paired-end sequencing was prepared using NEBNext High-Fidelity 2× PCR Master Mix (NEB) and a nested PCR approach as described in Schwartzman et al. 2016. Individual viewpoints are defined by US (upstream) and DS (downstream) primers within the DpnII fragment of interest (Supplementary Table 1). US and DS primers were designed with a melting temperature of 58 °C. DS primers were designed between 5–15 bp from the interrogated DpnII restriction site and containing the P5 sequence at their 5′ end. US primers were designed within a region of up to 100 bp of interrogated DpnII restriction site and with only minimal overlap with DS primers. Up to 14 US and DS primers were pooled for the multiplex PCR reaction, respectively. First PCR reaction: 50 μl reaction, containing 1x NEBNext High-Fidelity 2× PCR Master Mix, 0.3 μM US primer mix (each) and 0.3 μM TruSeq Primer 2.0 (P7), 200 ng DNA-bound on beads. PCR cycler setup: (1) 98 °C for 30 s, (2) 98 °C for 10 s, (3) 58 °C for 30 s, (4) 72 °C for 60 s, (5) Go to step 2 for 18 cycles in total, and (6) 72 °C for 5 min. For each sample, two PCR reactions were performed and then pooled for AMPure beads purification. A 1.2× volume of AMPure beads was added to the sample, mixed by pipetting, and incubated for 10 min at room temperature. Beads were separated on a magnet, and the clear supernatant was discarded. Beads were washed twice with 70% EtOH, air dried, and DNA was eluted in 30 μl water. Second PCR reaction: 50 μl reaction, containing 1× NEBNext High-Fidelity 2X PCR Master Mix, 0.3 μM DS primer mix (each) and 0.3 μM TruSeq Primer 2.0 (P7), 100 ng DNA from first PCR. PCR cycler setup: Corresponds to the setup of first PCR but with 15 cycles. For each sample 3–5 PCR reactions were performed and then pooled for size selection for fragments between 200 and 700 bp, using AMPure beads. 0.575× volume of AMPure beads were added to the sample, mixed by pipetting, and incubated for 10 min at room temperature. Beads were separated on a magnet, and the clear supernatant was transferred to a fresh tube. 0.3× volume of fresh AMPure beads was added to the supernatant, mixed, and incubated for 10 min at room temperature. Beads were separated on a magnet, and the clear supernatant was discarded. Beads were washed twice with 70% EtOH and air-dried for 5 min. DNA was eluted in 300 μl water. Libraries were multiplexed and sequenced using DNBseq technology to produce 50 bp paired-end reads and approximately 1–5 million raw sequencing read pairs for each viewpoint and genotype.

For the UMI-4C data analysis, raw fastq files were processed using the R package umi4cpackage 0.0.0.9000 (https://github.com/tanaylab/umi4cpackage). Contact profiles and domainograms, including difference plots, were generated using the default parameters and a minimum win_cov of 10.

**Virtual Hi-C (vHi-C) data generation and analysis.** The 3D chromatin models representing the *ptch2* and HoxD loci using the different 4C-seq datasets generated on WT and *ctcf*⁻/⁻ embryos were built using 4 Cin[58] (https://github.com/batxes/4Cin). The modeled loci are comprised within chromosome 2: 33,877,000–34,236,000 and chromosome 9: 1,628,000–2,591,500 of danRer10 zebrafish genome, respectively. Default parameters of the program were used except for the number of fragments that each bead represents, which was set to 2 for *ptch2* and 5 for HoxD.

**Statistical analyses.** For comparison of data distribution, two-tailed Wilcoxon's rank-sum tests or Student's *t* tests were used. Box plots represent the center line, median; box limits, upper and lower quartiles; whiskers, 1.5× interquartile range; notches, 95% confidence interval of the median. The statistical significance of contingency tables was assessed using Fisher's exact test.

**Reporting summary**. Further information on research design is available in the Nature Research Reporting Summary linked to this article.

## Data availability
The data that support this study are available from the corresponding author upon reasonable request. The HiC, ChIPmentation, RNA-seq, ATAC-seq, and UMI-4C data generated in this study have been deposited in the Gene Expression Omnibus (GEO) database under accession code GSE156099. The public datasets used in this study are available in the GEO database under accession codes: GSE105013 and GSE133437. Data for TF binding motifs were obtained from the JASPAR database (http://jaspar.genereg.net/). Source data are provided with this paper.

## Code availability
Custom code used in this study is available at the Gitlab repository: https://gitlab.com/rdacemel/hic_ctcf-null; https://gitlab.com/rdacemel/pancreasregulome.

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

## Acknowledgements

We dedicate this study to the memory of our friend, mentor, and colleague, José Luis Gómez-Skarmeta. We thank C. Paliou and J. López-Ríos for critical reading of the paper; P. Martínez-García for bioinformatics advice; C. Bolt and L. Delisle from the Duboule lab for technical advice with the UMI-4C protocol; F. Rencillas-Targa for providing the zebrafish-specific CTCF antibody; the CABD Fish, Proteomics and Microscopy Facilities for technical assistance; and C3UPO for the HPC support. J.L.G.-S. received funding from the ERC (Grant Agreement No. 740041), the Spanish Ministerio de Economía y Competitividad (Grant No. BFU2016-74961-P) and the institutional grant Unidad de Excelencia María de Maeztu (MDM-2016-0687). J.T. was funded by a 2019 Leonardo Grant for Researchers and Cultural Creators, BBVA Foundation. I.I.-A. acknowledges support from the Federation of European Biochemical Societies (FEBS Long-Term Fellowship). M.F. was funded by the European Union's Horizon 2020 research and innovation program under the Marie Skłodowska-Curie grant agreement [#800396] and a Juan de la Cierva-Formación fellow from the Spanish Ministry of Science and Innovation (FJC2018-038233-I). JMS-P was funded by a postdoctoral fellowship from Junta de Andalucía (DOC_0512).

## Author contributions

M.F., J.M.S.-P., and J.L.G.-S. conceived and designed the project; E.C.-M., M.F., A.N., M.A.-C., and J.M.S.-P. performed the experiments; J.M.S.-P., M.F., I.I.-A., R.D.A., and J.J.T. analyzed the data; M.F., J.M.S.-P., and J.L.G.-S. wrote the original paper. M.F. and J.M.S.-P. prepared the revised paper.

## Competing interests

The authors declare no competing interests.
