## [Peer Review File · Nature Communications]

REVIEWER COMMENTS

Reviewer #1 (Remarks to the Author):

In this manuscript authors generate a zygotic mutation of CTCF in zebrafish and analyze the consequences on animal development, genome architecture, transcription and other epigenomic aspects. The manuscript is well written and the data stand of high quality. The conclusions appear supported by the data. These new observations will contribute moving the field forward. I do not think the manuscript needs more experiments and encourage its timely publication in Nature Communications. Below I explain some reservations I have regarding how authors interpret the transcriptional consequences of CTCF depletion, from their data as well as previous work, which can be addressed by clarifying the manuscript.

Major points

1. When citing previous work authors say that depletion of mammalian CTCF only causes modest transcriptional alterations, citing work from Chen et al. Nature 2019, Kubo et al 2017 Biorxiv and Nora et al 2017 Cell. Chen et al. reported 819 genes dysregulated in human preimplantation embryos. Kubo et al. reported fewer than 10 genes dysregulated after depletion of CTCF in mouse ESCs after 24-48hrs and 253 after 96. Of note this study also reported overall preservation of chromatin architecture, which is contrast with other depletion experiments, including the present manuscript. Nora et al. reported 370, 1,353 and 4996 dysregulated genes in mouse ES cells after 1, 2 or 4 days of CTCF depletion respectively. These last numbers, especially at the later timepoints, are not in stark contrast with what authors observe here (718 after 24h and 6054 after 48h).

While I agree that previous studies showed a more limited immediate effect of gene regulation, some studies did report that loss of CTCF can ultimately affect the expression of many genes. While the expression changes at the late time points could be secondary effects of CTCF depletion (as is true at 48 hpf), the results in zebrafish are not a striking contrast to results in cell lines. Furthermore, mutation of the cohesin-interacting domain of CTCF in human HAP1 cells, which alters chromosome folding, leads to the dysregulation of more than 2000 genes (Li et al. Nature, PMID 31905366).

Given this I do not think authors can fairly say "an impacting result that contrasts to previous observations...", line 146-149. While I agree that the present study reports stronger transcriptional defects than what was observed after acute depletion in mammalian cells, I suggest authors rephrase their manuscript throughout to better reflect previous observations. I do not think doing so would diminish the impact of the present study.

2. Line 161: The authors suggest two explanations for why genes are differentially expressed in the CTCF mutant: CTCF binding directly to the TSS and changes in chromosome folding. A third possibility is that changes are secondary effects, downstream of genes that are directly regulated by CTCF. For example, if misregulation of a small number of genes arrests development, developmental genes that normally turn on at 48hpf may never be induced. The authors should acknowledge this possibility throughout the manuscript.

3. It remains to be investigated whether all functions of CTCF in transcriptional regulation are mediated by its role in loop extrusion and TAD folding. The CTCF KO presented here does not allow the authors to disentangle this, as all functions of CTCF are lost. Therefore, I do not think authors can say in their abstract "Our results demonstrate that CTCF and topologically associating domains are essential to regulate gene expression during embryonic development". I strongly suggest removing "and topologically associating domains" from this sentence. Same for in the discussion.

4. The claim that compartmentalization is decreased in the *ctcf* mutant is not clearly supported by the figures. The saddle plot in S1f seems to show little compartmentalization in the WT and no change in the *ctcf* mutant. Can the active and inactive regions be defined in a way that captures the plaid compartmentalization pattern clearly visible in panel e? For the figure S1g legend, please be more explicit in explaining how the values were calculated from the 2nd eigenvector. It may be helpful to show separate

values for the individual replicates to make it clear how reproducible the small change is. Of note Wutz et al. Elife 2020 PMID 32065581 also reported changes in compartmentalization strength upon CTCF KD in human cells.

Minor points:

1. It may be interesting to analyze the orientation of the CTCF motifs at TSSs of differentially expressed genes. If there is a bias in the orientation, it would suggest that CTCF's function at the genes is related to its role in loop extrusion (which is orientation-dependent), as was observed in mammalian cells in Nora et al. 2017

2. Figure 4a and S8a: please add a scale indicating the genomic distance.

3. Line 195 "We confirmed this by analyzing CTCF binding to DARs at 48hpf and found that 17.5% of up-regulated but 53.5% of down-regulated peaks were bound by CTCF". Up- or down- regulated peaks sounds strange – do authors mean peaks that gain / loose ATAC-seq signal?

4. These articles are relevant and should be discussed if space permits: Soshnikova et al. 2010 Dev cell 21145498; Kubo et al. Biorxiv 2020 <https://www.biorxiv.org/content/10.1101/2020.03.21.001693v1>

5. If space permits it would be useful to display the number of regions analyzed in each category next the heatmaps of fig 3e

6. Perez-Rico et al. 2020 concluded that CTCF does not demarcate TAD boundaries in zebra fish. Could authors include a citation and briefly comment on what could account for the different interpretation?

7. When citing studies that demonstrated the role of CTCF in mammalian genome folding authors should also cite Wutz et al. EMBO J 2017 PMID 29217591

8. It would be useful to include normalized bigwig files for the RNA-seq, ATAC-seq and ChIP-seq as supplementary files directly on GEO

Best wishes,
Elphege Nora

Reviewer #2 (Remarks to the Author):

Franke et al study the role of the architectural protein CTCF during zebrafish development. Recent studies have suggested that the role of CTCF in gene regulation and by that token 3D genome organization is minimal. Studying CTCF depletion in mammals is hampered that CTCF null mice die before gastrulation. The authors show that zebrafish do not have this problem, I would assume because of maternal deposition of CTCF and the later onset of zygotic genome activation (in terms of cell cycles, not absolute time). The authors show that until 24 hours post fertilization (hpf) *ctcf*^{-/-} embryos develop like wild-type embryos and also on gene expression the effects are quite limited. However, at 48hpf the *ctcf*^{-/-} embryos acquire a number of severe defects. These defect coincide unsurprisingly with a general loss of loops and TADs as has been observed for mammalian cell line systems as well. The authors further characterize the embryos by performing ATACseq and find a loss of open chromatin regions enriched for CTCF motifs. Finally by perfering UMI-4C the authors show that CTCF loss leads to upregulation *ptch2* which is associated with a loss of CTCF-associated chromatin interactions.

The data the authors show is interesting and clearly confirms a role for CTCF in the regulation of developmental genes. However, at the end of the day we are still left with the question how CTCF actually regulates these genes. Because at the TSS CTCF seems to be associated with gene activation, however, for *ptch2* CTCF loops seems to be associated with gene repression. This is quite an interesting observation, but

it is unclear to me how this would work. Are there any enhancers that are normally blocked by CTCF? It is unfortunate that the opportunity for more profound understanding of CTCF has been missed.

I see two major issues with the manuscript:

1. The effects the authors are seeing are likely indirect, and although the authors acknowledge this, the wording they choose is at often times ambiguous. Also some of the conclusions they become so vague that they become almost meaningless. I would suggest to make it absolutely clear that many of these effects are indirect.

For instance: "Interestingly, we observed that down-regulated genes that are enriched in developmental functions were mainly those without CTCF bound at their TSSs (Supplementary Fig. 4), raising the possibility that developmental genes could be de-regulated indirectly due to defects in chromatin folding."

What does this mean? Loss of CTCF leads to a loss of 3D genome structure, which then leads to a change in expression. It is a possibility, but equally likely (I would say much more likely) is that loss of CTCF leads to a decrease in the expression of key regulators which in turn cannot activate expression of developmental genes anymore. Although I am convinced that there are genes that are regulated through a change in the 3D genome/TAD structure if the authors want claim this they should do a systematic analysis of this which is currently lacking. Note that the manuscript is littered with these ambiguous but unnecessary claims.

2. Because the authors are looking in whole embryos the effects that they observe are necessarily heterogeneous. For instance, certain cells may lose CTCF earlier than others due to different protein stability between cells and different starting levels (this can actually be seen in figure 1b). This makes interpretation of the data much more complicated, because the RNAseq/ATACseq will be a mix of different cell types. To better disentangle the direct from the indirect effect the authors could choose to isolate a specific cell-type. Alternatively, they could perform single-cell RNAseq to see how CTCF loss affects expression at the single cell level (although I realize this may be an expensive exercise). However, their in situ hybridizations clearly show the problem with the bulk analysis.

Other points:

Figure 1f: please also show the enrichment of genomic background. In the text it is mentioned that 97% of TAD borders contain a CTCF site, however, without any genomic reference this number is meaningless.

Figure 2e: please specify the orientation of the CTCF motif at the TSS. Nora et al have shown that this orientation is specific for downregulated genes.

Reviewer #3 (Remarks to the Author):

Authors of Franke et al. investigate the in vivo function of CTCF during zebrafish development. They generated a mutant zebrafish strain using CRISPR targeting of *ctcf* and they then examined changes in higher order chromatin structure, gene expression, and chromatin accessibility, in zygotic null fish embryos. They conclude that CTCF is necessary for maintaining TADs and for regulation of developmental gene expression patterns. They highlight how prior in vitro studies are conflicting with regards to CTCF function in higher order structure and gene regulation state. In general this is a good study that I find quite interesting and timely. However, I have several major concerns which diminish my enthusiasm. They are detailed below.

Major concerns:

1. Loss of zygotic CTCF has not been effectively quantified and authors have not eliminated potential side-effects of truncated CTCF functioning as a dominant negative. Presumably, the mutation in *ctcf* would cause either a truncated protein, or complete loss of CTCF due to NMD of mRNA, but the authors do not distinguish

these. If a truncated protein is present in the embryos, it might function as a dominant negative, and compromise downstream analyses. Authors need to investigate this with additional experimentation (e.g. Western blots). If necessary, WT mRNA rescue experiments might be warranted. The authors should also quantify maternal CTCF protein to better justify the timing of the mutant phenotype. Does maternal CTCF compensate for zygotic loss at earlier timepoints? If so, authors should provide evidence.

2. Loss of CTCF has a considerable impact on global chromatin accessibility. More than 18,000 sites loss accessibility, but it is unclear whether this reduction is caused by primary vs. secondary effects. Nearly 9000 sites that decrease are not normally bound by CTCF. What causes these decreases? Are particular motifs enriched at these sites? Are these sites located proximal or distal to gene promoters? Do all CTCF motifs lose accessibility? What portion of CTCF motifs or binding sites change upon CTCF loss?

3. Loss of CTCF impacts only a subset of TADs. 2438 TADs are present in WT and 1178 in *ctcf* mutant, approximately half the number remaining in the absence of CTCF. Simply relying on these numbers, it is not possible to assess which TADs are lost, and which are gained in the CTCF mutant. If new TADs form in the absence of CTCF this should be investigated. Do new TADs form proximal to sites lacking a CTCF motif? Are there other factors that might be involved in higher order chromatin structure formation once CTCF is no longer present?

4. Impacts on gene expression may not be a direct consequence CTCF loss. In figure 1, authors demonstrate that CTCF levels are very low at 24hpf, but in figure 2, the major impacts on gene expression are not apparent until 48hpf, 24 hours after they demonstrate there to be CTCF loss. Similar impact occur at 24hpf in the ATAC study in Figure 3. Why does it take so long for gene expression patterns and chromatin accessibility to be impacted? Are TADs stable at 24hpf despite the absence of CTCF? If TADs and accessibility are stable, what is stabilizing them? Authors should investigate candidate chromatin contacts at 24hpf (perhaps using 3C/4C) to determine how stable these contacts are in the absence of CTCF. Authors should also discuss what factors might be involved in maintaining chromatin state in the absence of CTCF, and speculate as to why the impact of CTCF is delayed.

Minor Concerns.

1. Authors should provide evidence the p53 Morpholino injections were successful and not the result of experimental failure. Demonstrating a p53 mutant phenotype would be sufficient.

2. In figure 3i, authors should normalize peak numbers based on TAD size. The expectation is that larger TADS will have more peaks, perhaps causing this results independent of CTCF loss, and compromising the interpretation of this figure.

3. Authors should expand figure 4 to include additional examples, including gene clusters which one would expect be impacted by CTCF loss - such as the Hox clusters. Authors provide in situ measurements of *Hoxa5* and *Hoxa9* in Sup Fig 8. Looking at impacts of the *HoxA* locus similar to the data presented in Fig 4 would likely serve as a good example.

REVIEWER COMMENTS

Reviewer #1 (Remarks to the Author):

In this manuscript authors generate a zygotic mutation of CTCF in zebrafish and analyze the consequences on animal development, genome architecture, transcription and other epigenomic aspects. The manuscript is well written and the data stand of high quality. The conclusions appear supported by the data. These new observations will contribute moving the field forward. I do not think the manuscript needs more experiments and encourage its timely publication in Nature Communications. Below I explain some reservations I have regarding how authors interpret the transcriptional consequences of CTCF depletion, from their data as well as previous work, which can be addressed by clarifying the manuscript.

Major points

1. When citing previous work authors say that depletion of mammalian CTCF only causes modest transcriptional alterations, citing work from Chen et al. Nature 2019, Kubo et al 2017 Biorxiv and Nora et al 2017 Cell. Chen et al. reported 819 genes dysregulated in human preimplantation embryos. Kubo et al. reported fewer than 10 genes dysregulated after depletion of CTCF in mouse ESCs after 24-48hrs and 253 after 96. Of note this study also reported overall preservation of chromatin architecture, which is contrast with other depletion experiments, including the present manuscript. Nora et al. reported 370, 1,353 and 4996 dysregulated genes in mouse ES cells after 1, 2 or 4 days of CTCF depletion respectively. These last numbers, especially at the later timepoints, are not in stark contrast with what authors observe here (718 after 24h and 6054 after 48h).

While I agree that previous studies showed a more limited immediate effect of gene regulation, some studies did report that loss of CTCF can ultimately affect the expression of many genes. While the expression changes at the late time points could be secondary effects of CTCF depletion (as is true at 48 hpf), the results in zebrafish are not a striking contrast to results in cell lines. Furthermore, mutation of the cohesin-interacting domain of CTCF in human HAP1 cells, which alters chromosome folding, leads to the dysregulation of more than 2000 genes (Li et al. Nature, PMID 31905366).

Given this I do not think authors can fairly say “an impacting result that contrasts to previous observations...”, line 146-149. While I agree that the present study reports stronger transcriptional defects than what was observed after acute depletion in mammalian cells, I suggest authors rephrase their manuscript throughout to better reflect previous observations. I do not think doing so would diminish the impact of the present study.

We thank Elphege Nora for his review and constructive comments on the manuscript. We agree with his assessment that the total number of differentially expressed genes is not

dramatically different to some previous reports (especially Nora et al., 2017) and that this fact should not be a main message from our study. We have rephrased this throughout the manuscript and instead we now bring the focus on regulation of developmental genes and the consequences on embryo development that can be assessed by our animal model. See for instance:

Page 6, lines 180-183: “Compared to *in vitro* CTCF depletion approaches^{19,24,25}, our results reveal a considerable impact on developmental gene expression and indicate that CTCF is required for the expression of developmental genes during zebrafish embryogenesis.

2. Line 161: The authors suggest two explanations for why genes are differentially expressed in the CTCF mutant: CTCF binding directly to the TSS and changes in chromosome folding. A third possibility is that changes are secondary effects, downstream of genes that are directly regulated by CTCF. For example, if misregulation of a small number of genes arrests development, developmental genes that normally turn on at 48hpf may never be induced. The authors should acknowledge this possibility throughout the manuscript.

We agree that transcriptional regulation by CTCF could affect indirectly the expression of downstream genes and thank the reviewer for raising this point. To address the extent of these indirect effects, we have reanalyzed our ATAC-seq and RNA-seq data using a method based on footprinting analyses (Bentsen et al., Nat Commun 2020). This allowed us to determine which transcription factors show differential binding in the absence of CTCF, defining a CTCF transcription factor network and their putative target genes (new Supplementary Figure 11). We found that less than a half of the differentially expressed genes in our mutant could be explained by the downstream transcription factor network, indicating that the indirect effects due to miss-regulation of CTCF targets explain only a subset of those transcriptional alterations.

Page 10, lines 306-321: “To quantify the extent of such indirect effects, we leveraged our ATAC-seq data to perform differential TF binding analysis using a recently reported approach based on TF DNA footprints⁵⁴. As expected, we detected the CTCF motif with most reduced footprint signature, indicating reduced chromatin binding, but also 26 other motifs with altered footprints, 9 with increased and 15 with decreased footprints (**Supplementary Fig. 11a-d**). Next, we built a TF network based on the presence of TF footprints at TF gene promoters, using CTCF as starting point and the motifs with differential footprints. This defined a CTCF TF network including 24 of the 26 motifs with differential footprints, which corresponded to 38 zebrafish orthologous genes (**Supplementary Fig. 11e**). Although 17 of them were miss-regulated in *ctcf*^{-/-} embryos, our RNA-seq detected 452 further TF genes differentially expressed, indicating that only a small subset of miss-expressed TF genes could be explained by this CTCF network. Furthermore, the assignment of putative target genes to ATAC peaks containing these motifs identified only 47.9% of DEGs in our mutant (2,895 out of 6,049 genes) (**Supplementary Fig. 11f**). This analysis suggests that the role of CTCF as transcriptional regulator and the potentially associated

downstream TFs binding to DARs can explain a subset of the DEGs in *ctcf* knockout mutants”.

3. It remains to be investigated whether all functions of CTCF in transcriptional regulation are mediated by its role in loop extrusion and TAD folding. The CTCF KO presented here does not allow the authors to disentangle this, as all functions of CTCF are lost. Therefore, I do not think authors can say in their abstract “Our results demonstrate that CTCF and topologically associating domains are essential to regulate gene expression during embryonic development”. I strongly suggest removing “and topologically associating domains” from this sentence. Same for in the discussion.

We agree that the *ctcf* knockout mutant does not allow to separate both functions of CTCF. Where appropriate, we rephrased general conclusions about the function of CTCF. To better address changes in gene expression that are associated with changes in chromatin structure, we included promoter-centered HiChIP experiment and analysis, now included in new Figure 3. Considering this distinction, the rearranged abstract and introduction include the following statements.

Page 2, line 35-38: “Here we link the loss of CTCF and gene regulation during patterning and organogenesis in a *ctcf* knockout zebrafish model. CTCF absence leads to loss of chromatin structure and affects the expression of thousands of genes, including many developmental regulators.”

Page 3, lines 75-77: “Our results demonstrate that CTCF is essential to regulate gene expression during embryonic development at multiple levels, including the constraining of enhancer-promoter interactions within developmental RLs.”

Thank you.

*4. The claim that compartmentalization is decreased in the *ctcf* mutant is not clearly supported by the figures. The saddle plot in S1f seems to show little compartmentalization in the WT and no change in the *ctcf* mutant. Can the active and inactive regions be defined in a way that captures the plaid compartmentalization pattern clearly visible in panel e? For the figure S1g legend, please be more explicit in explaining how the values were calculated from the 2nd eigenvector. It may be helpful to show separate values for the individual replicates to make it clear how reproducible the small change is. Of note Wutz et al. Elife 2020 PMID 32065581 also reported changes in compartmentalization strength upon CTCF KD in human cells.*

We thank the reviewer for this comment. We have now improved the A/B compartment analysis by several implementations: (1) Using a resolution of 100 kb instead of 500 kb; (2) Excluding chromosome 4 from the enrichment analyses due to its extremely high repeat and heterochromatin content (Howe et al., Nature 2013); and (3) determining the A and B

compartments based on their enrichment in histone modifications, RNA and accessibility. During reanalysis we noticed that the assigned A and B compartment to the eigenvalues based solely on genome GC content were not consistent with those enrichments. A description for our methodology is now implemented in the method section:

Page 27, lines 980-989: “For determination of A and B compartments, 100-Kb binned HiC matrices were used. Pearson’s correlation matrices were calculated as previously described⁸⁸, using FAN-C⁸⁷. A and B compartments were determined using the values of the 2nd eigenvector, since the 1st eigenvector corresponded to chromosome arms in our experiments. The A compartment was assigned as the one with the highest enrichment in H3K27ac, H3K27me3, H3K4me3, RNA-seq and ATAC-seq signals. A/B domains were defined as consecutive regions with the same eigenvector sign. Enrichment profiles were calculated by ranking A and B compartments in fifty percentiles according to their eigenvalues and plotting their average observed/expected contact values. Interactions among A and B compartments were quantified separately for biological replicates at 500-Kb resolution.”

We also included new HiC data for 24 hpf and reanalyzed the level of compartmentalization using individual HiC replicates. Altogether, the new analysis (now implemented in Supplementary Figure 3) shows that there is a slight but consistent decrease in compartmentalization in the *ctcf* mutant that is similar to that shown by previous CTCF depletion studies (Wutz et al., 2020). The citation for Wutz et al., 2020 is included.

Page 5-6 lines 146-153: “Next, we analyzed A and B compartments in wild-type and *ctcf*^{-/-} embryos and, although we found a similar distribution of compartments, we detected a less intense plaid pattern of the Pearson’s correlation matrices in *ctcf*^{-/-} embryos both at 24 and 48 hpf (**Supplementary Fig. 3a**). Saddle plots of the interaction enrichment of genomic bins sorted by their eigenvalues detected a slight but consistent decrease in compartmentalization in mutant embryos that affected specifically interactions between active regions (**Supplementary Fig. 3b-d**). This is in agreement with recent CTCF depletion experiments in mammalian cells⁴⁷ and suggests that CTCF may be required for higher order chromatin structure.”

Minor points:

1. It may be interesting to analyze the orientation of the CTCF motifs at TSSs of differentially expressed genes. If there is a bias in the orientation, it would suggest that CTCF’s function at the genes is related to its role in loop extrusion (which is orientation-dependent), as was observed in mammalian cells in Nora et al. 2017

We have analyzed the orientation of the CTCF motifs at the TSS of differentially expressed genes, as suggested, and included the data as new Figure 2g. However, and in contrast to previous observations (Nora et al., 2017), we have not found a bias towards orientation in the same direction than transcription from the TSS, suggesting that CTCF function at these genes

may be more related with direct transcriptional regulation rather than a loop extrusion mechanism.

Page 7 lines 192-194: “In contrast to previous data¹⁹, we did not detect any clear bias in the orientation of the CTCF motif at TSS relative to transcription (**Fig. 2g**).”

2. *Figure 4a and S8a: please add a scale indicating the genomic distance.*

We revised our Figures to make sure scale bars are added to better appreciate genomic distances. Thank you.

3. *Line 195 “We confirmed this by analyzing CTCF binding to DARs at 48hpf and found that 17.5% of up-regulated but 53.5% of down-regulated peaks were bound by CTCF”. Up- or down-regulated peaks sounds strange – do authors mean peaks that gain / loose ATAC-seq signal?*

We have substituted the terms ‘up-regulated’ and ‘down-regulated’ by ‘increased’ or ‘decreased’, respectively, when referring to differentially accessible regions. Thank you for bringing this to our attention.

4. *These articles are relevant and should be discussed if space permits: Soshnikova et al. 2010 Dev cell 21145498; Kubo et al. Biorxiv 2020 <https://www.biorxiv.org/content/10.1101/2020.03.21.001693v1>*

We have discussed and implemented Soshnikova et al., Dev. Cell 2010, as well as the recently published Kubo et al., Nat Struct. Mol. Biol. 2021.

Soshnikova et al: Page 3, line 59-62: “In fact, due to the essential function of CTCF during cell cycle and the early embryonic lethality in mice³⁶⁻³⁹, our understanding of its function *in vivo* during organogenesis is limited to a few physiological contexts.”. Page 3, line 351-353: “The expression of *ptch2* in the pectoral fin buds was not detected likely due to their severely impaired development at this developmental stage (**Fig. 5b**), corroborating previous results in *ctcf*-deficient mouse limb buds³⁹.”.

Kubo et al: Cited at multiple occasions in the manuscript. In the introduction, results and discussion, especially when discussing our HiChIP results.

5. *If space permits it would be useful to display the number of regions analyzed in each category next the heatmaps of fig 3e*

We have added the number of regions analyzed for heatmaps of former Figure 3e (new Figure 4e), as well as other heatmaps and aggregate analyses when required. Thank you.

6. *Perez-Rico et al. 2020 concluded that CTCF does not demarcate TAD boundaries in zebra fish. Could authors include a citation and briefly comment on what could account for the different interpretation?*

We now cite the work of Pérez-Rico et al., 2020. Furthermore, we reanalyzed their ChIP-seq data in our new Supplementary Figure 4a when describing CTCF enrichment at TAD boundaries. We cross-verified our CTCF enrichment at TAD boundaries by using HiC data from Kaaij et al., 2018, which was originally used in Pérez-Rico et al., 2020. As with our HiC data, we found CTCF enrichment at TAD boundaries from Kaaij et al., while no CTCF enrichment was detected with the ChIP-seq from Perez-Rico et al., neither within our TAD boundaries nor within TAD boundaries called in HiC from Kaaij et al.

In addition, our reanalysis and comparison by motif enrichment revealed CTCF as top enriched motif in our CTCF ChIP-seq data (CTCF motif, Rank: 1, p-value: 1e-11,281, targets: 50.7%) whereas the CTCF-HA ChIP-seq data from Pérez-Rico et al. showed extremely low enrichment (CTCF-Like motif, Rank: 62, p-value: 1e-28, targets: 3.9%). This low motif enrichment in Pérez-Rico et al. could suggest un-specificity and might be related to the different chromatin pulldown strategies. In our study, we used an antibody directed against endogenous zebrafish CTCF (Carmona-Aldana et al., 2018) while Pérez-Rico et al. 2020 used HA antibody against endogenously HA-tagged CTCF. We cannot for certainty clarify this, but we implemented the reanalysis and our observation in the manuscript (now Supplementary Fig. 4a) to highlight the discrepancy with Perez-Rico et al 2020.

Pages 6, lines 159-163: “This enrichment was also observed using previously published HiC data of 24-hpf zebrafish embryos⁴³, but not with recently reported ChIP-seq data using an HA-tagged CTCF protein in zebrafish⁵⁰ (**Supplementary Fig. 4d**). In contrast to the latter study, we detected a clear CTCF motif prevalence and a high enrichment of CTCF at TAD boundaries in zebrafish, using an antibody against endogenous CTCF⁵¹.”

7. *When citing studies that demonstrated the role of CTCF in mammalian genome folding authors should also cite Wutz et al. EMBO J 2017 PMID 29217591*

Thanks, we now cite Wutz et al., 2017 in the introduction when referring to CTCF and genome folding. Page 3 lines 48-51:

8. *It would be useful to include normalized bigwig files for the RNA-seq, ATAC-seq and ChIP-seq as supplementary files directly on GEO*

We have included the requested normalized bigwig files on GEO under accession GSE156099.

Best wishes,

Elphege Nora

Reviewer #2 (Remarks to the Author):

*Franke et al study the role of the architectural protein CTCF during zebrafish development. Recent studies have suggested that the role of CTCF in gene regulation and by that token 3D genome organization is minimal. Studying CTCF depletion in mammals is hampered that CTCF null mice die before gastrulation. The authors show that zebrafish do not have this problem, I would assume because of maternal deposition of CTCF and the later onset of zygotic genome activation (in terms of cell cycles, not absolute time). The authors show that until 24 hours post fertilization (hpf) *ctcf*^{-/-} embryos develop like wild-type embryos and also on gene expression the effects are quite limited. However, at 48hpf the *ctcf*^{-/-} embryos acquire a number of severe defects. These defect coincide unsurprisingly with a general loss of loops and TADs as has been observed for mammalian cell line systems as well. The authors further characterize the embryos by performing ATACseq and find a loss of open chromatin regions enriched for CTCF motifs. Finally by performing UMI-4C the authors show that CTCF loss leads to upregulation *ptch2* which is associated with a loss of CTCF-associated chromatin interactions.*

*The data the authors show is interesting and clearly confirms a role for CTCF in the regulation of developmental genes. However, at the end of the day we are still left with the question how CTCF actually regulates these genes. Because at the TSS CTCF seems to be associated with gene activation, however, for *ptch2* CTCF loops seems to be associated with gene repression. This is quite an interesting observation, but it is unclear to me how this would work. Are there any enhancers that are normally blocked by CTCF? It is unfortunate that the opportunity for more profound understanding of CTCF has been missed.*

We thank the reviewer for the critical revision of the manuscript. In this revised version, we address the direct and indirect consequences of CTCF loss over gene expression using several approaches that aim to clarify the contribution of both CTCF functions in gene regulation during embryonic development (please, see below).

We agree that the mechanism of *ptch2* up-regulation in the *ctcf* mutant is quite intriguing. Unfortunately, we are not able at this stage to pinpoint changes in chromatin looping from the *ptch2* promoter that could easily explain this observation, such as ectopic interaction with a validated candidate enhancer. However, we extended our analysis at the locus by implementing additional UMI-4C viewpoints in wild-type and mutant embryos at 48 hpf, now included in Supplementary Figure 13a. Comparisons of individual viewpoints along the locus showed the described reduction of chromatin contacts within the *ptch2* regulatory landscape. In addition, we observed slight increase of contacts with regions outside the *ptch2* regulatory domain from multiple viewpoints. To investigate these changes more systematically we used all viewpoints for modelling distances at the locus. The new results are presented in new Figure 6 and Supplementary Figure 13 (main text page 12, lines 376-409). The analysis shows increased distances of the *ptch2* promoter with potential CREs

(ATAC peaks) within the *ptch2* domain and stronger intermingling with regions outside the *ptch2* regulatory domain. Although we cannot make conclusions about the miss-regulation of individual genes, we generally conclude that: page 12, lines 407-409: “We suggest that the loss of wild-type chromatin contacts within RLs and the potential intermingling of interactions across TADs contribute to the observed altered gene expression.”

I see two major issues with the manuscript:

1. The effects the authors are seeing are likely indirect, and although the authors acknowledge this, the wording they choose is at often times ambiguous. Also some of the conclusions they become so vague that they become almost meaningless. I would suggest to make it absolutely clear that many of these effects are indirect.

For instance: “Interestingly, we observed that down-regulated genes that are enriched in developmental functions were mainly those without CTCF bound at their TSSs (Supplementary Fig. 4), raising the possibility that developmental genes could be de-regulated indirectly due to defects in chromatin folding.”

What does this mean? Loss of CTCF leads to a loss of 3D genome structure, which then leads to a change in expression. It is a possibility, but equally likely (I would say much more likely) is that loss of CTCF leads to a decrease in the expression of key regulators which in turn cannot activate expression of developmental genes anymore. Although I am convinced that there are genes that are regulated through a change in the 3D genome/TAD structure if the authors want claim this they should do a systematic analysis of this which is currently lacking. Note that the manuscript is littered with these ambiguous but unnecessary claims.

We apologize for not making that part clear enough. We removed these ambiguous statements and distinguished in the revised manuscript between the mentioned indirect effects and genes that are regulated through a change in the 3D genome/TAD structure. The new manuscript provides a more systematic analysis by addressing the direct vs. indirect effects using two complementary approaches:

1- We have systematically analyzed the effects of CTCF loss over the chromatin contacts from active promoters by performing HiChIP of H3K4me3 at 48 hpf. Differential loop calling between WT and *ctcf* mutant embryos was performed using a recently developed tool, FitHiChIP (Bhattacharyya et al., Nat Commun 2019). We detected 677 loops that were altered in the mutant (new Figure 3a-b), consistent with recent data derived from mESCs (438 loops) and NPCs (915 loops) (Kubo et al., 2021). Differential loops with decreased contact frequency were mostly located within TADs, associated with CTCF binding and transcriptional down-regulation in the *ctcf* mutant. In contrast, loops with increased contact

frequency crossed more often TAD boundaries, were associated with increased expression and not with CTCF binding (new Figure 3c-h, page 7-8, lines 217-245). These results are consistent with a role of CTCF-mediated chromatin structure in restricting enhancer-promoter interactions within TADs to increase the contact frequency with enhancers located in the same TAD and preventing ectopic interaction with enhancers in the neighbor TADs.

The number of differential loops from promoters detected by this assay is limited, as previously mentioned (Kubo et al., 2021). This is in part due to the analysis being restricted to promoters with similar H3K4me3 levels in WT and *ctcf* mutant embryos, since it is not possible to distinguish changes in chromatin contacts from changes in promoter activation for those with significant differences in H3K4me3 levels. Therefore, additional genes might be affected by alterations in promoter contacts.

2- We have addressed the extent of indirect effects of CTCF function in transcriptional regulation by reanalyzing our ATAC-seq and RNA-seq data using a method based on footprinting analyses (Bentsen et al., Nat Commun 2020). This allowed us to determine which transcription factors show differential binding in the absence of CTCF, defining a CTCF transcription factor network and their putative target genes (new Supplementary Figure 11, main text page 10, lines 306-321). We found that less than half of the differentially expressed genes in our mutant could be explained by the downstream transcription factor network, indicating that the indirect effects due to miss-regulation of CTCF targets explain a subset of those transcriptional alterations.

2. Because the authors are looking in whole embryos the effects that they observe are necessarily heterogeneous. For instance, certain cells may lose CTCF earlier than others due to different protein stability between cells and different starting levels (this can actually be seen in figure 1b). This makes interpretation of the data much more complicated, because the RNAseq/ATACseq will be a mix of different cell types. To better disentangle the direct from the indirect effect the authors could choose to isolate a specific cell-type. Alternatively, they could perform single-cell RNAseq to see how CTCF loss affects expression at the single cell level (although I realize this may be an expensive exercise). However, their in situ hybridizations clearly show the problem with the bulk analysis.

We agree with the reviewer that isolating specific embryonic cell types or performing single cell RNA-seq and even multi-omics from multiple developmental time points in our mutant and WT siblings would be ideal to analyze CTCF loss in different cell types. However, the magnitude and complexity of these approaches, as well as available resources, is far above the scope of this manuscript. In the current manuscript, we want to establish the *ctcf* knockout in zebrafish as animal model to study the multifaceted function of CTCF and chromatin structure on gene regulation. However, we share the reviewers concern and in the revised manuscript, we include immunofluorescence assays with higher-resolution and from a time-series of five developmental stages monitoring the contribution of maternally provided CTCF protein in the *ctcf* mutant embryos, heterozygous and WT siblings. The immunofluorescent pictures of former Figure 1b were misleading as they showed background signals in the

mutants only. We apologize for this mistake. The new time-series and higher resolution of the generated data confirms the loss of CTCF protein at 24 hpf, with only background signal remaining in mutant embryos. Using earlier stages of development, we also show the gradual decay/loss of maternally provided CTCF protein, which is measurable at least until 18 somite stage in *ctcf* mutant embryos. CTCF protein quantification at the different stages and in different embryonic domains (anterior vs. posterior) showed the same dynamic and we did not observe measurable differences in CTCF protein stability in different embryonic domains or cell populations. This data is now implemented into new Figure 1b and Supplementary Figure 1d, page 4-5, lines 104-121.

Other points:

Figure 1f: please also show the enrichment of genomic background. In the text it is mentioned that 97% of TAD borders contain a CTCF site, however, without any genomic reference this number is meaningless.

We have now included a shuffle control for the enrichment of CTCF binding within TAD boundaries in former Figure 1f, which is now Figure 1e. Thank you.

Figure 2e: please specify the orientation of the CTCF motif at the TSS. Nora et al have shown that this orientation is specific for downregulated genes.

We have analyzed the orientation of the CTCF motif at the TSS of differentially expressed genes and included it as new Figure 2g. In contrast to previous observations (Nora et al., 2017), we did not find a bias towards orientation in the same direction than transcription from the TSS, suggesting that CTCF function at these genes may be more related with direct transcriptional regulation rather than a loop extrusion mechanism.

Page 7 lines 192-194: “In contrast to previous data¹⁹, we did not detect any clear bias in the orientation of the CTCF motif at TSS relative to transcription (**Fig. 2g**).”

Reviewer #3 (Remarks to the Author):

*Authors of Franke et al. investigate the in vivo function of CTCF during zebrafish development. They generated a mutant zebrafish strain using CRISPR targeting of *ctcf* and they then examined changes in higher order chromatin structure, gene expression, and chromatin accessibility, in zygotic null fish embryos. They conclude that CTCF is necessary for maintaining TADs and for regulation of developmental gene expression patterns. They highlight how prior in vitro studies are conflicting with regards to CTCF function in higher order structure and gene regulation state. In generally this is a good study that I find quite interesting and timely. However, I have several major concerns which diminish my enthusiasm. They are detailed below.*

We thank the reviewer for reviewing and comments on our manuscript.

Major concerns:

*1. Loss of zygotic CTCF has not been effectively quantified and authors have not eliminated potential side-effects of truncated CTCF functioning as a dominant negative. Presumably, the mutation in *ctcf* would cause either a truncated protein, or complete loss of CTCF due to NMD of mRNA, but the authors do not distinguish these. If a truncated protein is present in the embryos, it might function as a dominant negative, and compromise downstream analyses. Authors need to investigate this with additional experimentation (e.g. Western blots). If necessary, WT mRNA rescue experiments might be warranted. The authors should also quantify maternal CTCF protein to better justify the timing of the mutant phenotype. Does maternal CTCF compensate for zygotic loss at earlier timepoints? If so, authors should provide evidence.*

In the revised version, we have addressed both the maternal contribution of CTCF protein and the analysis of zygotic protein in wild type, heterozygous and homozygous embryos.

1. We have monitored the maternally provided CTCF protein by performing a time series of whole-embryo immunofluorescence assays using the zebrafish-specific CTCF antibody. This series covers blastula (1k-cell), gastrula (30% and 80% of epiboly), segmentation (18 somites) and pharyngula (24 hpf) stages of the zebrafish early development in WT, heterozygous *ctcf*^{+/-} and homozygous *ctcf*^{-/-} embryos. These experiments show a significant drop in CTCF protein levels in homozygous mutants from 80% of epiboly that reaches background levels at 24 hpf, while heterozygous mutants show similar or slightly decreased CTCF levels compared to the WT (new Figure 1b and new Supplementary Figure 1d, , page 4-5, lines 104-121). We observe a progressive decay in maternally provided CTCF protein that is consistent with the progressive effect that we observe between 24 and 48 hpf at the

level of chromatin architecture (see reviewer point 4), gene expression and chromatin accessibility.

2. We now address the consequences of our CTCF mutation at the protein level. By integrating these new data, we exclude a potential dominant negative effect of a CTCF truncated protein by the following three reasons:

(A) Western blot assays and protein quantification at 48-hpf embryos using an antibody recognizing the N-terminal domain of CTCF shows absence of wild-type CTCF protein in homozygous mutants and slight reduction in heterozygous embryos. We did not detect truncated CTCF protein in hetero- or homozygous mutants, suggesting mutant protein degradation. The data is now implemented in the main text, page 4, lines 92-96 and new Supplementary Figure 1a.

(B) Heterozygous *ctcf*^{+/-} animals develop normal and are indistinguishable from their wild-type siblings. We included pictures of all three genotypes derived from *ctcf*^{+/-} crosses in new Figure 1a.

(C) As described above, and thanks to the reviewer's suggestion of quantifying maternal CTCF protein, we are now able to better justify the timing of the mutant phenotype. The loss of maternal CTCF between 18 somite stage (18 hpf) and 24 hpf coincide with the onset of a 'molecular phenotype' (differential gene expression and chromatin accessibility) at 24 hpf and the subsequent onset of the mutant phenotype between 36 and 48 hpf.

2. Loss of CTCF has a considerable impact on global chromatin accessibility. More than 18,000 sites loss accessibility, but it is unclear whether this reduction is caused by primary vs. secondary effects. Nearly 9000 sites that decrease are not normally bound by CTCF. What causes these decreases? Are particular motifs enriched at these sites? Are these sites located proximal or distal to gene promoters? Do all CTCF motifs lose accessibility? What portion of CTCF motifs or binding sites change upon CTCF loss?

We argue that altered chromatin accessibility may be a consequence of differential transcription factor (TF) binding. These changes likely represent secondary effects of CTCF loss due to changes in the expression of transcription factor genes. We show in the revised manuscript that ATAC peaks with reduced accessibility in the *ctcf* mutants are enriched in DNA binding motifs of the zinc finger C2H2 family (that include CTCF) but also for motifs of the bHLH, HMG and homeobox families. Of those, ATAC peaks without CTCF binding tend to be located at larger distances to the nearest TSS (new Supplementary Figure 7d-e, main text page 9, lines 263-267). We identified a total of 24,398 CTCF binding sites (Supplementary Figure 1b) from which only 7,350 sites (30%) lose accessibility as shown in new Figure 4e.

To address whether changes in accessibility may be explained by changes in the expression of transcription factor genes, we analyzed the expression levels of TF families enriched in the differentially accessible regions (DARs). We considered all transcripts of TFs that belong

to the same TF family and that are expressed in zebrafish embryos at 48 hpf (according to curated data from the ZFIN database). We also used non-related families of DNA binding factors as negative control (new Supplementary Figure 10). The results of this analysis are implemented in the main text, page 10, line 300-306: “For TF families associated to increased DARs we found that only transcripts of the tp53 family were significantly up-regulated in *ctcf*^{-/-} embryos (**Supplementary Fig. 10**). In contrast, transcripts of all families associated to reduced DARs, including C2H2-type zinc fingers, bHLH, HMG, and homeobox, showed significantly reduced expression levels (**Supplementary Fig. 10**). Thus, the decreased accessibility at developmental CREs may be explained by a down-regulation of TF genes caused by CTCF absence, which in turn would lead indirectly to miss-regulation of their downstream target genes.”

*3. Loss of CTCF impacts only a subset of TADs. 2438 TADs are present in WT and 1178 in *ctcf* mutant, approximately half the number remaining in the absence of CTCF. Simply relying on these numbers, it is not possible to assess which TADs are lost, and which are gained in the CTCF mutant. If new TADs form in the absence of CTCF this should be investigated. Do new TADs form proximal to sites lacking a CTCF motif? Are there other factors that might be involved in higher order chromatin structure formation once CTCF is no longer present?*

We apologize for the confusion generated by referring to TADs detected in the mutant in the original manuscript. In our HiC analyses, we have computationally called TAD boundaries and from this inferred TADs. Since we detect less TAD boundaries in *ctcf* mutants, the remaining inferred TADs from persistent and new boundaries lead to the miss-interpretation that there are less but larger TADs in the mutant.

To avoid further miss-interpretation, we have revised this section and talk about TAD boundaries instead of TADs at page 5, lines 137-141. We have detected 2,582 and 2,416 TAD boundaries in WT embryos at 24 and 48 hpf, respectively, while only 1,666 and 1,157 are detected in *ctcf* mutant embryos. The subset of them that are mutant-specific show significantly lower overlap with CTCF binding sites, while the common boundaries (common with WT) are more likely to contain persistent CTCF binding sites (new Supplementary Figure 2f-g). In addition, analysis of the insulation score and RNA expression levels around boundaries suggest a lower insulation of mutant-specific boundaries, and higher transcription levels at both common and mutant-specific boundaries (new Supplementary Figure 2h, main text page 5, lines 141-145) suggests a role of transcription in sustaining boundaries in the absence of CTCF.

4. Impacts on gene expression may not be a direct consequence CTCF loss. In figure 1, authors demonstrate that CTCF levels are very low at 24hpf, but in figure 2, the major impacts on gene expression are not apparent until 48hpf, 24 hours after they demonstrate there to be CTCF loss. Similar impact occur at 24hpf in the ATAC study in Figure 3. Why

does it take so long for gene expression patterns and chromatin accessibility to be impacted? Are TADs stable at 24hpf despite the absence of CTCF? If TADs and accessibility are stable, what is stabilizing them? Authors should investigate candidate chromatin contacts at 24hpf (perhaps using 3C/4C) to determine how stable these contacts are in the absence of CTCF. Authors should also discuss what factors might be involved in maintaining chromatin state in the absence of CTCF, and speculate as to why the impact of CTCF is delayed.

We have performed new HiC experiments in WT and *ctcf* mutant embryos at 24 hpf (new Figure 1c-d, Supplementary Figures 2-4). These data reveal that chromatin structure is already impaired at this stage, consistent with the gradual decay in maternally provided CTCF protein at earlier stages and its loss at 24 hpf (new Figure 1b). However, the effect on chromatin structure, for example measured by insulation scores (new Figure 1d), is slightly lower at 24 hpf than at 48 hpf. This suggests complete loss of CTCF protein and associated chromatin structure starting around 24 hpf, which also coincided with the onset of changes in gene expression and chromatin accessibility at 24 hpf.

Later, in the transition between 24 and 48 hpf, zebrafish embryos progressively undergo lineage differentiation and genes that have to be induced or repressed during differentiation fail to do so, as suggested by the negative correlation of gene expression changes between 24 and 48 hpf in WT versus mutant embryos (new Supplementary Figure 5c-d, main text page 6, lines 182-186).

Another contributor to the increased gene expression defects observed at 48 hpf are indirect effects, induced by primary CTCF target genes that subsequently activate or repress their corresponding target genes. To address indirect effects induced by CTCF loss, we have reanalyzed our ATAC-seq and RNA-seq data using a method based on footprinting analyses (Bentsen et al., Nat Commun 2020). This allowed us to determine which transcription factors show differential binding in the absence of CTCF, defining a downstream transcription factor network and their putative target genes (new Supplementary Figure 11, main text page 10, lines 306-321). We found that almost half of the differentially expressed genes in our mutant could be explained by the downstream transcription factor network, indicating that the indirect effects due to miss-regulation of CTCF targets indeed contribute to the observed transcriptional alterations at 48 hpf.

Minor Concerns.

1. Authors should provide evidence the p53 Morpholino injections were successful and not the result of experimental failure. Demonstrating a p53 mutant phenotype would be sufficient.

We show in new Supplementary Figure 8b and d that the regions with higher accessibility in *ctcf* mutants injected with the p53 morpholino lose the enrichment of the p53 family DNA binding motif and decrease the expression of several well-known p53 target genes compared with non-injected embryos.

2. *In figure 3i, authors should normalize peak numbers based on TAD size. The expectation is that larger TADS will have more peaks, perhaps causing this results independent of CTCF loss, and compromising the interpretation of this figure.*

We have now normalized the number of ATAC peaks of former Figure 3i to the size of the TADs, as requested, and plotted it as number of peaks per Mb. The conclusion is still that TADs containing miss-regulated developmental genes are larger and with higher density of DARs. Thank you for bringing this to our attention.

3. *Authors should expand figure 4 to include additional examples, including gene clusters which one would expect be impacted by CTCF loss - such as the Hox clusters. Authors provide in situ measurements of Hoxa5 and Hoxa9 in Sup Fig 8. Looking at impacts of the HoxA locus similar to the data presented in Fig 4 would likely serve as a good example.*

We have expanded former Figure 4 including UMI-4C data from the HoxD cluster (former Supplementary Figure S8), since we have used these two loci for 3D modelling of distances (new Figure 6). In addition, we have performed the requested UMI-4C experiments using the *hoxa5a* and *hoxa9a* promoters as viewpoints. These data are included in new Supplementary Figure S12 (main text, page 11, lines 368-370) and shows that, similar to other loci (new Figure 5), CTCF loss leads to a decrease in chromatin interactions from these promoters that may explain the down-regulated expression of these genes shown by *in situ* hybridization.

REVIEWERS' COMMENTS

Reviewer #1 (Remarks to the Author):

Authors have added a lot of new data and addressed my suggestions. I encourage timely publication at this stage.

-For the description of the HiChIP results, it would be helpful to define what you mean by "loop." People often use "loop" to refer to peaks in Hi-C data, where interactions are enriched between two discrete loci compared to neighboring regions, but the examples here appear to be broader regions of high interactions.
-Please check the labels on Figure 6C and 6H. If they are single differential heatmaps comparing WT to CTCF mutant, there shouldn't be labels for the two different genotypes in the corners.

Reviewer #2 (Remarks to the Author):

Franke et al. have addressed my comments in their revision. They have also added additional experiments to strengthen the paper.

They have performed Hi-ChIP of H3K4me3 in order to identify promoter-enhancer loops. These type of experiments are always challenging, because a loss of H3K4me3 (for instance for a downregulated gene) will lead to an inability to detect a loop (even though the loop may still be there). One question I still have is whether the down-regulated TSSs that have a CTCF site (36% figure 2e) are also enriched in the class of loops with CTCF at both anchors that are lost in Figure 3e (loops with CTCF at both anchors, 32.2%).

Regarding this analysis they also write that 32% of lost loops have CTCF at both anchors, which is an increase over the stable and the increased, but can the authors speculate what happens at the 68% of loops that do not have CTCF at both anchors.

Other points:

Typos

* Figure 1e: "suffle" -> shuffle, Figure 2g "same than" -> same as.

Reviewer #3 (Remarks to the Author):

There was an impressive amount of additional results added to this manuscript during revision. Authors have nearly addressed all my concerns. My remaining concern deals with my prior point #4, where gene expression and accessibility defects do not arise at 24hpf, despite complete absence of CTCF. The additional HiC data helps to mitigate this concern, as chromatin contacts appear to be disrupted at 24hpf, but these new results lead to additional concerns. Presently the authors propose a model where gradual ctfc reduction leads to failed chromatin structure at 24hpf (Fig 1) but according to their ATAC-Seq data (Fig 4), fewer than 35 CTCF binding sites are disrupted. How do the authors reconcile this issue? Does a separate factor maintain accessibility at all other CTCF binding sites in the absence of CTCF? Does the process of closing chromatin at previously accessible sites occur gradually? If the later, this result would suggest that chromatin accessibility patterns (at 24hpf) do not regulate higher order structure (at 24hpf). Presumably this may occur because accessibility is less dynamic than structural changes, which is a very interesting result. This result also suggests that CTCF's function as a trans-looping factor may be more important/dynamic than its function as a DNA binding factor. The authors should provide substantial discussion of this issue in order to accommodate my final concern, and to provide the conceptual advance necessary for publication.

REVIEWERS' COMMENTS

Reviewer #1 (Remarks to the Author):

Authors have added a lot of new data and addressed my suggestions. I encourage timely publication at this stage.

Thank you for your helpful and constructive review of our manuscript.

-For the description of the HiChIP results, it would be helpful to define what you mean by "loop." People often use "loop" to refer to peaks in Hi-C data, where interactions are enriched between two discrete loci compared to neighboring regions, but the examples here appear to be broader regions of high interactions.

Although HiChIP signal results in broad regions of promoter interactions, loops between H3K4me3 peaks and discrete regions are called by FitHiChIP (peak-to-all mode). To avoid confusion with loops called in HiC data, we will refer to these loops as 'HiChIP loops'. We have stated this in the text:

Page 7 lines 208-209: "To avoid confusion with loops called in HiC data, we will refer to loops called in HiChIP data as 'HiChIP loops' hereafter."

-Please check the labels on Figure 6C and 6H. If they are single differential heatmaps comparing WT to CTCF mutant, there shouldn't be labels for the two different genotypes in the corners.

Thank you for bringing this mistake to our attention. We have now edited former Figure 6C and 6H (new Figure 8C and 8H) to remove wrong labels.

Reviewer #2 (Remarks to the Author):

Franke et al. have addressed my comments in their revision. They have also added additional experiments to strengthen the paper.

Thank you for your helpful and constructive review of our manuscript.

They have performed Hi-ChIP of H3K4me3 in order to identify promoter-enhancer loops. These type of experiments are always challenging, because a loss of H3K4me3 (for instance for a downregulated gene) will lead to a inability to detect a loop (even though the loop may still be there). One question I still have is whether the down-regulated TSSs that have a CTCF site (36% figure 2e) are also enriched in the class of loops with CTCF at both anchors that are lost in Figure 3e (loops with CTCF at both anchors, 32.2%).

This an interesting question. We have analyzed the overlap between increased and decreased promoter interactions with differentially expressed genes containing or not CTCF at their TSS. This analysis shows that decreased promoter interactions are enriched for down-regulated genes containing CTCF at their TSS (41.1% versus 27.8% of stable promoter interactions overlapping the same class of genes). We have included this data as new Supplementary Figure S6b.

Page 8 lines 236-239: “We found that increased loops were indeed significantly more associated with the TSS of up-regulated genes than stable loops, while decreased loops showed a higher overlap with the TSS of down-regulated genes (Figure 3f), and in particular with TSSs bound by CTCF (Supplementary Figure 6b).”

Regarding this analysis they also write that 32% of lost loops have CTCF at both anchors, which is an increase over the stable and the increased, but can the authors speculate what happens at the 68% of loops that do not have CTCF at both anchors.

Our results show that CTCF promotes intra-TAD interactions and prevents inter-TAD interactions. Thus, the reduction of intra-TAD interactions as a consequence of the loss of CTCF-mediated insulation would decrease the contact probability inside TADs and this would be independent on CTCF binding to particular interactions.

We have speculated this possibility in the text:

Page 8 lines 230-233: “Interestingly, 67.8% of decreased HiChIP loops did not show CTCF binding at both anchors, suggesting that the absence of CTCF-mediated insulation decreases the contact probability within TADs independently of CTCF binding.”

Other points:

Typos

* Figure 1e: “suffle” -> shuffle, Figure 2g “same than” -> same as.

We have corrected these mistakes, thank you.

Reviewer #3 (Remarks to the Author):

There was an impressive amount of additional results added to this manuscript during revision. Authors have nearly addressed all my concerns. My remaining concern deals with my prior point #4, where gene expression and accessibility defects do not arise at 24hpf, despite complete absence of CTCF. The additional HiC data helps to mitigate this concern, as chromatin contacts appear to be disrupted at 24hpf, but these new results lead to additional concerns. Presently the authors propose a model where gradual ctf reduction leads to failed chromatin structure at 24hpf (Fig 1) but according to their ATAC-Seq data (Fig 4), fewer than 35 CTCF binding sites are disrupted. How do the authors reconcile this issue? Does a separate factor maintain accessibility at all other CTCF binding sites in the absence of CTCF? Does the process of closing chromatin at previously accessible sites occur gradually? If the later, this result would suggest that chromatin accessibility patterns (at 24hpf) do not regulate higher order structure (at 24hpf). Presumably this may occur because accessibility is less dynamic than structural changes, which is a very interesting result. This result also suggests that CTCF's function as a trans-looping factor may be more important/dynamic than its function as a DNA binding factor. The authors should provide substantial discussion of this issue in order to accommodate my final concern, and to provide the conceptual advance necessary for publication.

Thank you for your helpful and constructive review of our manuscript.

We agree with the reviewer that our new results suggest that chromatin structure is more dynamic than accessibility, since the effect of CTCF loss is first observed at the level of chromatin structure and later at the level chromatin accessibility and gene expression. To assess whether the loss of chromatin accessibility observed at 48 hpf occurs gradually from 24 hpf, when CTCF is already absent, we have visualized the ATAC-seq signal at 24 hpf at the differentially accessible regions detected at 48 hpf. New heatmaps in Figure 4e demonstrates that sites losing accessibility at 48 hpf show also decreased accessibility at 24 hpf, although to a lower extent, and this occurs mainly at ATAC peaks bound by CTCF. This suggests that (1) CTCF function as an architectural protein is more dynamic than its function as a DNA binding factor, and (2) accessibility at regulatory elements not bound by CTCF occurs later likely as a consequence of decreased transcription factor expression and chromatin occupancy, as we show in Supplementary Figures 10 and 11.

We have discussed this point in the Results and Discussion sections:

Page 9 lines 273-276: “Interestingly, we found that reduced DARs at 48 hpf, especially those bound by CTCF, showed slightly reduced accessibility already at 24 hpf (Figure 4e). This suggests that the effect of CTCF loss on chromatin accessibility is progressive but less dynamic than the effect on chromatin structure at 24 hpf (Figure 1c-d).”

Page 15 lines 468-470: “Importantly, reduced chromatin accessibility and developmental gene expression arise later than defects in chromatin structure, suggesting that the latter is acutely affected by CTCF loss before changes in chromatin accessibility patterns arise.”

Thank you.